# Debiasing Attention Mechanism in Transformer without Demographics

**Shenyu Lu, Yipei Wang & Xiaoqian Wang** *
Elmore Family School of Electrical and Computer Engineering
Purdue University
West Lafayette, IN 47906, USA
`{lu876,wang4865,joywang}@purdue.edu`

## Abstract

Although transformers demonstrate impressive capabilities in a variety of tasks, the fairness issue remains a significant concern when deploying these models. Existing works to address fairness issues in transformers require sensitive labels (such as age, gender, etc.), which can raise privacy concerns or violate legal regulations. An alternative way is through fairness without demographics. However, existing works that improve Rawlsian Max-Min fairness may impose overly restrictive constraints. Other methods that use auxiliary networks could be parameter inefficient. In this paper, we present a new approach to debiasing transformers by leveraging their inherent structure. By reconsidering the roles of important components (queries, keys, and values) in the attention mechanism, we introduce a simple yet effective debiasing strategy from two perspectives: 1) Grounded in theoretical analysis, we normalize and apply absolute value operations to *queries* and *keys* to minimize the bias in attention weight allocation; 2) We reduce the bias within *values* through local alignment via contrastive learning. Throughout the entire process, our approach does not require any sensitive labels. Furthermore, to enhance memory efficiency in the training phase, we propose a strategy that debiases only the last encoder to improve fairness in pre-trained models. We conduct experiments in computer vision and natural language processing tasks and show that our method is comparable and even outperforms the state-of-the-art method with substantially lower energy consumption.

## 1 Introduction

Transformer-based models demonstrate immense power and achieve remarkable success in both computer vision and natural language processing fields. The transformer architecture is based on the concept of self-attention, which allows the model to weigh the importance of different tokens or patches of the input sequence when making predictions (Vaswani et al., 2017).

Fairness is a crucial factor to consider in machine learning models. For example, Gong et al. (2021) highlight the potential risk of applying biased face recognition systems in law enforcement. Although transformer models exhibit outstanding performance, Sudhakar et al. (2023), Qiang et al. (2023), and Baldini et al. (2021) observe that transformers make biased predictions in vision and NLP domains. Addressing fairness issues in transformers is a significant but challenging task. Our work focuses on mitigating biases within transformer architectures.

Existing works attempt to address fairness issues in transformers in two ways: targeted alignment for debiasing transformers (TADeT) and debiasing self-attention (DSA). Sudhakar et al. (2023) proposed to debias transformer by aligning "Query" in different sensitive groups within the same task. DSA (Qiang et al., 2023) generates adversarial examples by attacking spurious features, and aligns attention weight between training samples and adversarial examples. However, these methods necessitate sensitive attributes (such as gender and race) to improve fair transformers. Collecting such information is not only costly but may also raise privacy concerns.

Several studies (Hashimoto et al., 2018; Lahoti et al., 2020; Liu et al., 2021; Creager et al., 2021; Chai et al., 2022) propose to resolve general fairness issues without sensitive information.

---

*Corresponding author.

Hashimoto et al. (2018) propose to optimize the worst group to achieve equal utility in different sensitive groups. However, this method is likely to be affected by outliers, leading to a significant drop in performance. Lahoti et al. (2020) share the same idea to solve fairness issues but use a reweighting strategy. However, their method incorporates adversarial training, which introduces instability. Moreover, improving the worst-performing group imposes a strong constraint on the objective function. Liu et al. (2021) suggest a method where a network is trained twice, emphasizing samples misclassified by the initial model. However, this double-training approach lacks efficiency and runs counter to green deep learning initiatives (Xu et al., 2021). Creager et al. (2021) propose a method that is also a two-stage method, where the first step is to perform environment inference, followed by leveraging a robust optimization method to train the network. The limitation here is that their environment inference relies on the Bernoulli distribution, making it applicable only to binary sensitive labels. A recent study (Chai et al., 2022) tackles fairness without demographics via knowledge distillation. However, this approach involves a teacher network that is significantly larger than the student network for making predictions, thus being time-consuming and parameter-inefficient.

Additionally, large-scale pre-trained models also face fairness issues. Many existing methods require tuning all the parameters to achieve a balance between fairness and accuracy, but updating a large number of parameters can be prohibitively expensive (Petersen et al., 2021).

In summary, existing works of fair transformers require sensitive attributes. Existing methods for fairness without demographics either need a large auxiliary network or lay a strong constraint. For addressing fairness concerns in pre-trained models, tuning all parameters lacks efficiency and necessitates high-demand computational resources.

To resolve the above challenges, we propose a novel approach to tailoring the fairness considerations to the transformer encoder. Instead of debiasing the entire representation of the transformer encoder which places significant constraints on the representation, we propose to debias each component (Query, Key, and Value) in the attention formulation. This approach, derived from theoretical foundations, presents a simple yet effective strategy to address the fairness concern in transformers without the necessity of auxiliary networks or iterative training.

We debias from two perspectives within the attention mechanism.
- Attention allocation: Drawing inspiration from theoretical analyses, we find that normalizing and applying the absolute value to $\mathbf{q}$ and $\mathbf{k}$ reduces discrepancies in attention weight between different sensitive groups.

- Local alignment on value: We mitigate bias in $\mathbf{v}$ through local alignment. This is accomplished with a supervised contrastive learning approach, which encourages the core segments from different sensitive groups to be similar.

We conduct extensive experiments on real-world datasets, encompassing various classification tasks in computer vision and natural language processing (NLP) fields. Furthermore, we provide a GPU memory-efficient solution to address fairness in pre-trained models. We append a fairness-aware encoder on these models and only train that encoder. We demonstrate its efficacy in enhancing fairness within pre-trained models.

We summarize our contributions as follows:
- To the best of our knowledge, we are the first to enhance fairness in transformers without demographics.

- Our algorithm achieves effective results in both computer vision and NLP domains.

- Our method could plug in a pre-trained model to improve fairness without the need for re-training, resulting in improved memory efficiency.

## 2 RELATED WORK

### 2.1 FAIRNESS INTERVENTION

Existing fairness intervention strategies can be categorized as pre-processing, in-processing, and post-processing (Wang et al., 2024). Pre-processing methods mitigate biases by optimizing the data, including sampling (Qraitem et al., 2023; Kamiran & Calders, 2012; Chakraborty et al., 2020; Yao & Liu, 2023), transformation (Yao et al., 2022; Calmon et al., 2017), and augmentation (Zietlow et al., 2022; Jang et al., 2021; Wang et al., 2020; Cheong et al., 2022). Nonetheless, the methods discussed incur significant costs due to the necessity for human annotation of sensitive attributes (Chen et al.,

2024); Post-processing methods adjust outputs directly to meet fairness objectives, with various approaches such as flip prediction (Hardt et al., 2016), score transformation (Alghamdi et al., 2022; Wei et al., 2021), threshold adjustment (Jang et al., 2022; Xian et al., 2023); In-processing methods are integrated into the model design and training phase, fostering the development of fair models (Wan et al., 2023). These can be classified into fair regularizer (Agarwal et al., 2018; Kamishima et al., 2011; Zemel et al., 2013; Jiang et al., 2020; Zafar et al., 2019; Lowy et al., 2021), adversarial learning (Zhang et al., 2018; Wadsworth et al., 2018), disentanglement (Sarhan et al., 2020; Locatello et al., 2019; Creager et al., 2019) an so on. In this work, we adopt an in-processing strategy because it offers more flexibility (Song et al., 2024) and allows us to directly address fairness issues during the training process. Compared to existing in-processing methods, our approach leverages the unique structure of the transformer and incorporates a tailored design to specifically address fairness concerns within these architectures.

## 2.2 FAIRNESS WITHOUT DEMOGRAPHICS

Distributionally robust optimization (DRO) (Hashimoto et al., 2018) achieves Rawlsian Max-Min fairness by optimizing the risk of the worst-case samples. Adversarially reweighted learning (ARL) (Lahoti et al., 2020) aims to optimize the performance of computationally identifiable samples by reweighting them, thus improving the worst-case performance. Knowledge distillation (KD) (Chai et al., 2022) leverages the soft labels generated by a teacher network and combines them with hard labels to debias a student model. Just Train Twice (JTT) (Liu et al., 2021) follows a reweighting paradigm, emphasizing error samples in the training set by giving them elevated weights. The training phase is split into two stages: an Empirical Risk Minimization (ERM) stage, succeeded by a reweighted ERM stage. Environment Inference for Invariant Learning (EIIL) (Creager et al., 2021) is also a two-stage method. The initial step employs a trained network to infer group labels, and the second step utilizes a robust optimization strategy to re-train the network. Learning from Failure (LfF) (Nam et al., 2020) involves training two neural networks concurrently: one with amplified bias and another focusing on samples that contradict this bias. Through this process, the second network is conditioned to correct the errors made by the first.

## 2.3 FAIRNESS-AWARE TRANSFORMERS

Recently, Sudhakar et al. (2023) and Qiang et al. (2023) aim to solve fairness issues in vision transformers. Sudhakar et al. (2023) proposed to use an adversarial training technique to hide the sensitive information in the classification token. Plus, they apply $\mathcal{L}_2$ Loss to penalize query vectors with large discrepancies among different sensitive groups with the same target label. However, debiasing on query may not be sufficient for resolving fairness issues in transformers, and the introduced adversarial training may incur instability issues. Qiang et al. (2023) propose a two-stage method to mitigate bias in vision transformer (ViT). They first identify sensitive related patches and use adversarial attacks to perturb those patches. Next, they train a ViT using both the original training set and the attacked dataset to perform attention weight alignment. However, this work hypothesizes that the sensitive attribute must be identifiable and has no overlap with target related patch. This laid a too strong constraint and may have a negative effect on the model performance. Moreover, both existing methods require sensitive attributes during the training phase.

## 2.4 CONTRASTIVE LOSS

The contrastive loss has been proven to be more robust (Xue et al., 2022) and has a better generalization performance than cross-entropy loss (Khosla et al., 2020). Supervised contrastive loss (Supcon) (Khosla et al., 2020) utilizes label information to pull together normalized embeddings of the same class and push apart normalized embeddings from different classes. Differing from self-supervised representation learning (Chen et al., 2020; Tian et al., 2020), SupCon selects positive samples from the same class within a mini-batch, leading to many positive and negative samples for each anchor, which enhances performance.

## 3 METHOD

**Problem setup** We denote a dataset $\mathcal{D} : (X, Y, A) \in \mathcal{X} \times \mathcal{Y} \times \mathcal{A}$, with $X$ as features, $Y$ being target labels, and $A$ being sensitive attributes. Our goal is to learn a classifier $f_\theta : X \mapsto \hat{Y}$, where $\hat{Y}$ is a prediction labels that satisfies fairness criteria. For example, Demographic parity (DP) requires $\hat{Y} \perp A$, Equal opportunity (EOp) requires $\hat{Y} \perp A | Y = 1$, and Equalized odds (EOd) requires $\hat{Y} \perp A | Y = y, \ y \in \{0, 1\}$. We focus on the setting where we do not have access to sensitive attributes in the training set.

**Background and notation** The informative capability of the transformer model can be attributed to its attention mechanism. In a transformer, an input image (or a sequence in NLP) is divided into

patches (tokens in NLP). These patches or tokens are first processed through an embedding layer. Subsequently, they undergo a transformation in the encoder through three linear modules, which map them to query, key, and value (Vaswani et al., 2017; Dosovitskiy et al., 2020). In this work, we denote $\mathbf{Q} \in \mathbb{R}^{N \times d_k}$ as query, $\mathbf{K} \in \mathbb{R}^{N \times d_k}$ as key, and $\mathbf{V} \in \mathbb{R}^{N \times d_v}$ as value, where $N$ is the number of tokens. Additionally, we denote $\mathbf{q} \in \mathbb{R}^{d_k}$ as a query vector corresponding to a patch, extracted from $\mathbf{Q}$. Similarly, we denote $\mathbf{k} \in \mathbb{R}^{d_k}$ as a key vector, $\mathbf{v} \in \mathbb{R}^{d_v}$ as a value vector. We use the subscript "cls" as a special symbol to denote a patch or token used for a downstream task. This notation aligns with the conventions established in (Devlin et al., 2018).

**Motivation**  A recent study (Sudhakar et al., 2023) reveals disparities in average query across different sensitive groups when performing the same task, highlighting discriminatory behavior in the attention mechanism. In addition, Sudhakar et al. (2023); Qiang et al. (2023) underscore the importance of the attention mechanism in mitigating bias issues. They empirically validate that addressing fairness concerns within the attention mechanism is effective and yields good performance compared to other in-processing methods. Based on these findings, our work focuses on debiasing within the attention mechanism. To this end, we revisit the attention mechanism (Vaswani et al., 2017):

$$\text{Attention}(\mathbf{Q}, \mathbf{K}, \mathbf{V}) = \text{softmax}(\frac{\mathbf{Q}\mathbf{K}^\top}{\sqrt{d_k}})\mathbf{V}, \tag{1}$$

The attention mechanism in equation 1 can be viewed as a two-stage operation: 1) *Attention allocation* performed by the inner product of $\mathbf{Q}$ and $\mathbf{K}$, and 2) *Weighted sum* over $\mathbf{V}$. We attribute the discrimination in a transformer-based model to 1) a misallocation of attention weight and 2) biased representation of $\mathbf{V}$. Based on the observations, we present a two-stage approach that does not require sensitive attributes to mitigate discrimination in a transformer encoder.

### 3.1 FAIRNESS-AWARE ATTENTION WEIGHT RELOCATION

Drawing inspiration from the principle of minimizing the statistical distance between representations of various groups to enhance fairness (Balunović et al., 2021), we introduce a method that aligns the expected attention weights across diverse demographic groups without accessing sensitive attributes. To be specific, we denote $w = \frac{\mathbf{q}^\top \mathbf{k}}{\sqrt{d_k}}$ as attention weight of two tokens (patches). Given the fairness consideration, our optimization problem can be written as:

$$\min |\delta| \quad s.t. \quad \delta = \mathbb{E}[w|a = 0] - \mathbb{E}[w|a = 1] \tag{2}$$

which is to minimize the disparity in attention weights $w$ between different sensitive groups. Directly solving equation 2 can be challenging due to the inaccessibility of the sensitive attribute. We present Theorem 1, which shows that a series operation can effectively reduce the attention allocation disparity among different sensitive groups.

**Assumption 1.** *Denote $P(\mathbf{q}|a = i) = \mathcal{N}(\boldsymbol{\mu}_{q,i}, \Sigma_{q,i})$, $P(\mathbf{k}|a = i) = \mathcal{N}(\mu_{k,i}, \boldsymbol{\Sigma}_{k,i})$, $i \in \{0, 1\}$. For $q_j \in \mathbf{q}$ and $k_j \in \mathbf{k}$, $q_j$ and $k_j$ are independent[1], $j \in \{1, ..., d_k\}$. The base rate for $P(a = 0) = \lambda_0$ and $P(a = 1) = \lambda_1$. The distribution of query $\mathbf{q}$ and the key $\mathbf{k}$ of a transformer can be written as:*

$$\mathbf{q} \sim \lambda_0 \mathcal{N}(\boldsymbol{\mu}_{q,0}, \Sigma_{q,0}) + \lambda_1 \mathcal{N}(\boldsymbol{\mu}_{q,1}, \Sigma_{q,1}) \tag{3}$$

$$\mathbf{k} \sim \lambda_0 \mathcal{N}(\boldsymbol{\mu}_{k,0}, \Sigma_{k,0}) + \lambda_1 \mathcal{N}(\boldsymbol{\mu}_{k,1}, \Sigma_{k,1}) \tag{4}$$

For a transformer, the expected difference in attention weight is

$$|\delta| = |\mathbb{E}[\frac{\mathbf{q}^\top \mathbf{k}}{\sqrt{d_k}}|a = 0] - \mathbb{E}[\frac{\mathbf{q}^\top \mathbf{k}}{\sqrt{d_k}}|a = 1]| = \frac{1}{\sqrt{d_k}}|\sum_{j=1}^{d_k}(\mathbb{E}[q_j k_j|a = 0] - \mathbb{E}[q_j k_j|a = 1])|$$

$$= \frac{1}{\sqrt{d_k}}|\sum_{j=1}^{d_k}(\mu_{q,0,j}\mu_{k,0,j} - \mu_{q,1,j}\mu_{k,1,j})| \tag{5}$$

$\delta$ in a vanilla transformer is *unbounded*. To resolve this issue, we reference Theorem 1.

**Theorem 1.** *For a vector $\mathbf{x} = [x_1, ..x_d] \in \mathbb{R}^d$, define $n(\mathbf{x}) = [\frac{x_1 - \mathbb{E}[x_1]}{\sqrt{Var[x_1]}}, ..., \frac{x_d - \mathbb{E}[x_d]}{\sqrt{Var[x_d]}}]$, and $m(\mathbf{x}) = [|x_1|, ..., |x_d|]$. Given query $\mathbf{q} \in \mathbb{R}^{d_k}$ and key $\mathbf{k} \in \mathbb{R}^{d_k}$, denote the debiased query and key as $\mathbf{q}^{de} = m \circ n(\mathbf{q}) \in \mathbb{R}^{d_k}$, $\mathbf{k}^{de} = m \circ n(\mathbf{k}) \in \mathbb{R}^{d_k}$. Under Assumption 1, the disparity in debiased attention weight is upper bounded by*

---

[1]The independence of $q_j$ and $k_j$ follows the same assumption presented in (Vaswani et al., 2017).

$$|\delta^{de}| = \left| \mathbb{E}[\frac{\mathbf{q}^{de\top}\mathbf{k}^{de}}{\sqrt{d_k}}|a=0] - \mathbb{E}[\frac{\mathbf{q}^{de\top}\mathbf{k}^{de}}{\sqrt{d_k}}|a=1] \right| \leq \sqrt{d_k}\left[ (\sqrt{\frac{2}{\pi\lambda_0}}+\sqrt{\frac{\lambda_1}{\lambda_0}})^2 + (\sqrt{\frac{2}{\pi\lambda_1}}+\sqrt{\frac{\lambda_0}{\lambda_1}})^2 \right]$$

For the attention mechanism in a transformer encoder, as shown in equation 5, the disparity of the attention weights across different demographic groups has the potential to diverge toward infinity. Theorem 1 indicates that upon normalization of vectors in $\mathbf{Q}$ and $\mathbf{K}$, and taking their absolute values, the disparity in attention allocation could be bounded at a constant, wherein the constant incorporates the base rate of the training set and the embedding dimension. Note that when the sensitive attribution is balanced (e.g. The numbers of male and female samples are the same in CelebA), the disparity is minimal. We leave the details of the proof for Theorem 1 in the appendix.

In practice, we compute attention weight based on $\mathbf{q}^{de}$ and $\mathbf{k}^{de}$, where $\mathbf{q}^{de} = m \circ n(\mathbf{q}) \in \mathbb{R}^{d_k}$, $\mathbf{k}^{de} = m \circ n(\mathbf{k}) \in \mathbb{R}^{d_k}$, as stated in Theorem 1. During the inference period, we use the mean and standard deviation estimated from the training set to normalize $\mathbf{q}$ and $\mathbf{k}$ for the test samples. To estimate these statistics, we utilize the running mean ($\mathbf{e}$) and running standard deviation ($\mathbf{s}$) as suggested by (Ioffe & Szegedy, 2015). The running mean is initialized with zero tensors, and the running standard deviation is initialized with one tensors. We update the running mean and running standard deviation using a momentum-based approach:

$$\mathbf{e} \leftarrow (1-p)\mathbf{e} + p\mathbf{e}^{\text{norm}}, \quad \mathbf{s} \leftarrow (1-p)\mathbf{s} + p\mathbf{s}^{\text{norm}}, \tag{6}$$

where $p$ is the momentum. we set $p = 0.1$ by following existing protocols (Ba et al., 2016; Ioffe & Szegedy, 2015). $\mathbf{e}^{\text{norm}}$ and $\mathbf{s}^{\text{norm}}$ are sample mean and sample standard deviation in a batch.

## 3.2 LOCAL ALIGNMENT ON VALUE

In transformer structures, the value of a token (a patch) can be considered as a local representation. Tokens (patches) with high attention weights encapsulate core objects, ensuring target-specific information encoding. For considerations of fairness, it is crucial that the encoding within these tokens (patches) remains consistent. Conversely, tokens (patches) with low attention weights, typically representative of background elements, enhance the richness of the final representation.

In representation learning, Chen et al. (2020) demonstrates that using a nonlinear projection head improves the representation quality. We follow this finding and define a small nonlinear head $g : \mathbf{v} \mapsto \mathbf{z}$ to map value $\mathbf{v}$ to a latent representation $\mathbf{z}$. To mitigate bias in the representation while preserving the discriminative power, we aim to ensure similar representations within the same target label, while dissimilar representations from different target labels, regardless of the sensitive attribute. A natural choice for achieving this objective is through supervised contrastive loss (Khosla et al., 2020):

$$\mathcal{L}_{\text{alignment}} = \sum_i \frac{-1}{|P(i)|} \sum_{p \in P(i)} \log \frac{\exp(\mathbf{z}_i \cdot \mathbf{z}_p/\tau)}{\sum_{k \in B(i)} \exp(\mathbf{z}_i \cdot \mathbf{z}_k/\tau)}, \tag{7}$$

where $i \in I = \{1, ..., M\}$ is the index of anchor, $M$ is the number of samples in a mini-batch, $B(i) = I\backslash\{i\}$, $P(i) = \{p \in B(i) : y_p = y_i\}$ is the set of indices of all positive samples in a mini-batch. $\tau$ is a temperature hyperparameter, we set $\tau = 0.07$. We selected the Supervised Contrastive Loss in accordance with our objective of achieving consistent representations. Though the recently proposed Fair Supervised Contrastive Loss (FSCL) integrates fairness by specifying negative samples within an identical sensitive group (Park et al., 2022). FSCL might compromise the efficacy of contrastive learning and necessitates the inclusion of both target and sensitive labels.

We argue that directly employing supervised contrastive learning on the representation of the entire image or sentence does not necessarily contribute to fairness. Park et al. (2022) demonstrate that with the use of Supervised Contrastive Learning (SupCon) to pre-train a ResNet model, issues of unfairness still persist. We attribute this phenomenon to the correlation between the target and sensitive labels. Since the network contrasts the entire representation, it can still leverage sensitive-related information. From this observation, we propose an attention-weight guided local value alignment technique to effectively tackle fairness concerns in the representation. The learning framework contains the following steps:

- We define $w_i = \frac{\mathbf{q}_{\text{cls}}^{de} \cdot \mathbf{k}_i^{de}}{\sqrt{d_k}}$ as the attention weight of the $i$-th token (patch), $W = \{w_1, ..., w_l\}$, where $l$ is the number of patches (in vision) or the number of tokens (in NLP). We select $\mathbf{v}_{s^*}$ with top $t$ attention weights: $\mathbf{s}^* = \arg\max_{\mathbf{s}} \sum_{i \in \mathbf{s}} w_i, \; s.t. \; w_i \in W, |\mathbf{s}^*| = t$

- We concatenate all $\mathbf{v}_{s*}$ and apply a nonlinear projection head $g(\cdot)$ to map from the value vector to representations: $\mathbf{z} = g(\mathbf{v}_{s*})$. We then apply supervised contrastive loss equation 7 on $\mathbf{z}$ to perform the local alignment of the value vector.

**Why does local alignment help fairness:** Our method debiases on the attention mechanism using two steps. The first step of debiasing the attention weight (in section 3.1) is crucial for the second step (in section 3.2) when selecting patches with high attention weights. Utilizing these debiased weights from the first step, we select the top $t$ patches based on target label relevance. Incorporating supervised contrastive learning encourages patches within a class to share similar representations, focusing on encoding information related to the target label, disregarding sensitive information. This approach reduces the sensitive information encoded in the representations, thereby enhancing fairness. It's important to note that without debiasing the attention weights in the first step, the technique of local alignment on value in the second step does not contribute to fairness. This is evidenced by our ablation study in Appendix F, where using local alignment alone barely impacts fairness. In contrast, combining debiased attention with local alignment achieves optimal results in terms of fairness.

### 3.3 LEARNING FRAMEWORK

We present our learning framework in Figure 1. Our learning framework contains two steps: First, we normalize each $\mathbf{q}$ and $\mathbf{k}$ and take their absolute values in the last encoder layer to compute attention weights. This approach ensures that attention allocation is less affected by sensitive attributes. Next, we select value $\mathbf{v}$ associated with top $t$ attention weights to perform local value alignment. Throughout this process, we do not leverage sensitive attributes. We summarized our method in an algorithm, detailed in Appendix G.

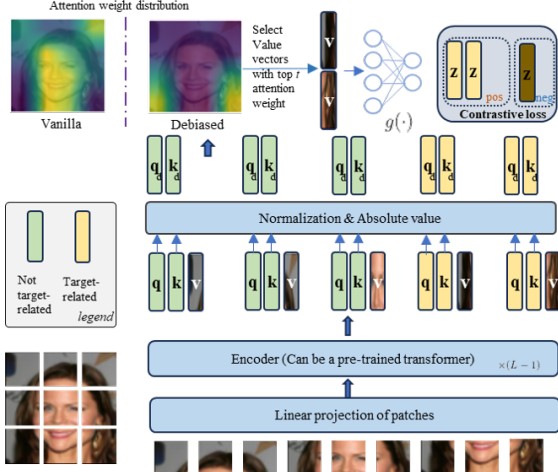

Figure 1: Our learning framework is illustrated using the vision transformer applied to the CelebA dataset, specifically for $y =$ blond hair, $a =$ male. An input image is processed through $L - 1$ transformer encoder layers. In the $L$-th layer, for each image patch, we normalize and take the absolute value of the query vectors and key vectors to debias the attention weights. Subsequently, we select the value vectors corresponding to the top $t$ debiased attention weights, concatenate these $t$ vectors, and map them to a new representation $\mathbf{z}$ using the function $g(\cdot)$. We apply a supervised contrastive loss to this representation $\mathbf{z}$ for contrastive learning.

### 3.4 GPU MEMORY EFFICIENT COMPUTATION

**Last encoder training** Even if the pre-trained model is powerful, fairness issues still exist. However, mitigating bias in large pre-trained models is challenging. Fine-tuning all the parameters is not only time-consuming but also heavily relies on high-performance computational devices.

The bias is spread in all the layers of encoders. However, as demonstrated by (Kirichenko et al., 2022), the objective of empirical risk minimization (ERM) training is sufficient for learning the core features on which a debiasing procedure can be deployed at the very last layer of the network.

Based on this observation, we append our proposed encoder layer to the top of the large pre-trained model. The pre-trained model is capable of capturing the useful features within an image (Caron et al., 2021) or sentence (Clark et al., 2019). We employ the pre-trained model as a feature extractor

and only train our fairness-aware one-layer encoder. This method does not require a GPU with large memory capacity, because parameters in pre-trained models are not updated. We show it has maintained utility and improved fairness in section 4.3.

# 4    EXPERIMENTS

We evaluate our method across various settings. We first explore our method in CV by conducting experiments on two real-world datasets. We next extend the application of our method to the NLP field. Additionally, we test the capability of our method to address fairness issues in pre-trained models with limited resources. Finally, we explore the trade-off between fairness and accuracy. To reproduce our experiment, we have made the code available at `https://github.com/lu876/Debiasing-Attention-Mechanism-in-Transformer-without-Demographics`.

## 4.1    VISION EXPERIMENTS

**Experimental setup**    We test all methods on two real-world datasets: CelebA (Liu et al., 2015), and UTK (Zhang & Qi, 2017). For CelebA dataset, we follow the fairness study's protocol and take $y =$ Blond Hair, $a =$ Male (Sagawa et al., 2019). UTK dataset is a widely used facial dataset in fairness research. In this study, we follow the task outlined in (Hong & Yang, 2021). The task is to predict $y =$ gender, the sensitive attributes are $a \in$ {White, not White}.

We compare our method with **ERM** and other state-of-the-art methods without demographics: **DRO** (Hashimoto et al., 2018), **ARL** (Lahoti et al., 2020), **KD** (Chai et al., 2022), **JTT** (Liu et al., 2021), and **LfF** (Nam et al., 2020). Please refer to the Appendix for detailed implementation information. In our implementation (**Ours**), the backbone is the same as the ERM model. We normalize and apply the absolute value to $\mathbf{Q}$ and $\mathbf{K}$ in the last encoder layer. We select $\mathbf{V}$ with the 2 highest attention weights to perform the local alignment on value procedure. For vision tasks, all methods are trained from scratch. We evaluate each method w.r.t accuracy and fairness. For the fairness metric, we use demographic parity (DP), Equal Opportunity (EOp), and equalized odd (EOd), we leave details on the computation of fair metrics in the appendix.

We train all methods on a single NVIDIA RTX-3090 GPU with 24576 MiB memory. Each method is independently trained three times, and we report the mean and standard deviation of the results. We follow a protocol for efficiency assessment in machine learning and report the overall energy consumption, which can be computed by (Wang et al., 2023):

$$E = \int_0^H u(t)p_0 \mathrm{d}t \quad (\text{Wh}), \tag{8}$$

where $u(t) \in [0, 1]$ (in percent) is the GPU utilization level, $H$ represents the duration of a training process lasting $H$ hours on a GPU. $p_0$ is the power requirement for the GPU. For an NVIDIA RTX-3090 GPU, $p_0 = 350$ W. In practice, we sample $u(t)$ per minute.

**Vision experimental results**    Results from experiments on the UTK and CelebA datasets can be found in Tables 1 and 2. We mark the best results in dark blue, and second-best in light blue. In the UTK dataset, our method achieves enhanced fairness performance with a slight decrease in accuracy. In the CelebA dataset, we observe that LfF attains optimal fairness. However, this comes at the expense of a significant reduction in accuracy, where the base rate for $y = 0$ is 84.67%, and the accuracy of LfF is merely 87.97%. JTT demonstrates a balanced performance between fairness and utility. However, in a smaller-scale dataset, the performance of JTT approaches that of ERM training. We hypothesize that this is due to JTT's tendency to overfit the up-weighted training set. Our proposed method not only yields outcomes that are comparable to those of JTT, but also effectively mitigates bias existing in smaller datasets. Remarkably, our method requires significantly lower energy consumption compared to JTT. We provide a visualization of attention weight in the last encoder layer, please refer to the appendix.

|  | ERM | DRO | ARL | KD | JTT | LfF | Ours |
|---|---|---|---|---|---|---|---|
| DP ↓ | $11.55 \pm 0.87$ | $11.73 \pm 0.76$ | $13.54 \pm 0.31$ | $9.89 \pm 1.02$ | $11.29 \pm 1.42$ | $11.26 \pm 1.07$ | $\mathbf{9.72 \pm 0.50}$ |
| EOp ↓ | $10.49 \pm 1.38$ | $10.47 \pm 1.01$ | $11.94 \pm 0.30$ | $8.69 \pm 1.26$ | $9.19 \pm 0.59$ | $11.55 \pm 0.44$ | $\mathbf{8.46 \pm 0.46}$ |
| EOd ↓ | $9.50 \pm 0.92$ | $9.66 \pm 0.71$ | $11.72 \pm 0.37$ | $7.97 \pm 0.93$ | $9.31 \pm 1.35$ | $9.39 \pm 0.86$ | $\mathbf{7.76 \pm 0.41}$ |
| ACC ↑ | $\mathbf{84.26 \pm 0.43}$ | $83.37 \pm 0.66$ | $80.27 \pm 1.53$ | $81.33 \pm 0.65$ | $83.68 \pm 0.48$ | $80.58 \pm 1.03$ | $83.11 \pm 1.38$ |
| $E$(Wh) ↓ | 26.54 | 12.05 | 34.77 | 55.94 | 62.20 | 27.48 | 27.65 |

Table 1: Classification results in terms of accuracy (ACC), fairness (DP, EOp, EOd) and energy consumption (E(Wh)) on UTK dataset: sensitive attribute $a =$ race, label $y =$ gender.

| | ERM | DRO | ARL | KD | JTT | LfF | Ours |
|---|---|---|---|---|---|---|---|
| DP↓ | $16.92 \pm 0.55$ | $16.88 \pm 0.33$ | $\mathbf{12.74 \pm 3.46}$ | $17.44 \pm 0.59$ | $18.04 \pm 0.32$ | $19.95 \pm 3.80$ | $16.34 \pm 1.94$ |
| EOp↓ | $42.82 \pm 0.47$ | $40.21 \pm 1.94$ | $40.55 \pm 4.38$ | $41.93 \pm 0.47$ | $36.17 \pm 1.08$ | $\mathbf{31.79 \pm 9.91}$ | $38.01 \pm 1.32$ |
| EOd↓ | $22.90 \pm 0.10$ | $22.57 \pm 0.58$ | $21.32 \pm 2.65$ | $22.54 \pm 0.40$ | $20.26 \pm 1.72$ | $\mathbf{19.26 \pm 3.55}$ | $20.43 \pm 0.68$ |
| ACC↑ | $94.26 \pm 0.06$ | $94.06 \pm 0.02$ | $92.75 \pm 1.34$ | $\mathbf{94.37 \pm 0.28}$ | $92.75 \pm 0.02$ | $87.97 \pm 4.08$ | $94.06 \pm 0.43$ |
| $E$(Wh)↓ | 144.86 | 141.75 | 251.71 | 287.58 | 936.24 | 611.10 | 172.18 |

Table 2: Classification results on CelebA dataset: $a$ = male, $y$ = blond hair.

## 4.2 NLP EXPERIMENTS

**Expermental setup** We evaluate our method within the domain of natural language processing (NLP) utilizing both the HateXplain (Mathew et al., 2021) and MultiNLI (Williams et al., 2017) datasets. For the HateXplain dataset, the target label, denoted by $y$, is equal to 0 for normal sentences and 1 for those that are considered hateful or offensive. In line with (Baldini et al., 2021) for choosing sensitive attributes, we assess whether a given corpus contains information that pertains to the attribute of gender ($a$). For the MultiNLI dataset, we follow (Liu et al., 2021), the task is to predict the entailment of the two sentences, $y$={entailed, neutral, contradictory}. The sensitive labels are $a$={no negation, negation}.

Pre-trained models have significantly advanced the field of NLP. To ensure an equitable comparison, we leverage a pre-trained model and augment it with an additional encoder. Specifically, we select "BERT Large" (Devlin et al., 2018) and "BERT Base" (Devlin et al., 2018) to represent large-scale (340M parameters) and medium-scale (110M parameters) models, respectively. For NLP tasks, we exclude ARL, KD, and LfF from the comparison because these methods necessitate an auxiliary network, making it challenging to align with the specifications of a pre-trained model. We apply AdamW to optimize the parameters. The learning rates of $10^{-5}$ are used for both "BERT Large"and "BERT Base". All methods share the same batch size and optimizer configuration. In the baseline model, the add-on layer is the same as one of the encoder layers of the backbone model.

**NLP experimental results** Table 3 presents results for the HateXplain dataset. It can be observed that our model not only achieves optimal fairness but also retains utility without compromising performance. Remarkably, when the backbone is the BERT base model, our method outperforms other methods in fairness and utility. Given the relatively small size of the HateXplain dataset, JTT tends to overfit, leading to suboptimal performance. This observation aligns with the results observed in the CV task.

Table 4 presents the results for the MultiNLI dataset. Our method outperforms both the ERM and DRO approaches. However, it falls short in comparison to JTT for fairness. This can be attributed to the base rates of the two groups in MultiNLI, which are $\lambda_0 = 92.9\%$ and $\lambda_1 = 7.1\%$. According to Theorem 1, the upper bound on the disparity in the expected attention allocation is proportional to $\frac{1}{\lambda_0} + \frac{1}{\lambda_1}$. Consequently, a highly skewed base rate results in an elevated upper bound.

| | ERM (BL) | DRO | JTT | Ours | ERM (BB) | DRO | JTT | Ours |
|---|---|---|---|---|---|---|---|---|
| DP↓ | $12.38 \pm 2.78$ | $10.45 \pm 0.35$ | $\mathbf{3.32 \pm 3.00}$ | $11.65 \pm 0.91$ | $8.88 \pm 0.84$ | $8.31 \pm 1.87$ | $\mathbf{6.42 \pm 3.16}$ | $11.30 \pm 1.97$ |
| EOp↓ | $5.43 \pm 3.56$ | $6.40 \pm 1.10$ | $13.12 \pm 3.59$ | $\mathbf{5.31 \pm 1.38}$ | $8.55 \pm 1.27$ | $7.80 \pm 2.54$ | $9.18 \pm 3.93$ | $\mathbf{4.74 \pm 3.28}$ |
| EOd↓ | $5.37 \pm 1.75$ | $5.53 \pm 0.63$ | $7.55 \pm 0.95$ | $\mathbf{3.51 \pm 1.08}$ | $6.62 \pm 0.44$ | $5.65 \pm 1.72$ | $5.41 \pm 1.53$ | $\mathbf{3.40 \pm 0.97}$ |
| ACC↑ | $\mathbf{79.69 \pm 0.15}$ | $79.21 \pm 0.26$ | $76.25 \pm 0.07$ | $79.61 \pm 0.32$ | $77.74 \pm 0.43$ | $77.56 \pm 0.44$ | $77.21 \pm 1.19$ | $\mathbf{77.89 \pm 0.78}$ |
| $E$(Wh)↓ | 124.32 | 144.14 | 265.48 | 142.51 | 22.58 | 36.87 | 59.75 | 34.00 |

Table 3: Fine-tuning on HateXplain dataset. (BL denotes "BERT Large", BB denotes "'BERT base')

| | ERM (BL) | DRO | JTT | Ours | ERM (BB) | DRO | JTT | Ours |
|---|---|---|---|---|---|---|---|---|
| DP↓ | $47.76 \pm 0.07$ | $46.19 \pm 1.34$ | $\mathbf{43.88 \pm 0.21}$ | $44.63 \pm 2.43$ | $47.35 \pm 0.97$ | $48.72 \pm 0.71$ | $\mathbf{44.27 \pm 0.23}$ | $47.25 \pm 0.22$ |
| EOp↓ | $14.71 \pm 1.40$ | $12.81 \pm 0.86$ | $\mathbf{4.62 \pm 0.01}$ | $10.36 \pm 0.16$ | $18.03 \pm 2.23$ | $17.47 \pm 2.82$ | $\mathbf{8.98 \pm 1.14}$ | $13.86 \pm 0.43$ |
| EOd↓ | $10.81 \pm 0.02$ | $9.95 \pm 0.95$ | $\mathbf{5.53 \pm 0.47}$ | $8.59 \pm 0.34$ | $13.52 \pm 1.30$ | $14.11 \pm 1.05$ | $\mathbf{8.35 \pm 0.64}$ | $12.67 \pm 0.25$ |
| ACC↑ | $\mathbf{84.33 \pm 0.02}$ | $83.33 \pm 0.60$ | $82.70 \pm 0.08$ | $83.95 \pm 0.02$ | $81.48 \pm 0.53$ | $\mathbf{81.63 \pm 0.12}$ | $76.75 \pm 0.60$ | $81.36 \pm 0.01$ |
| $E$(Wh)↓ | 1827.56 | 1791.07 | 5417.70 | 1934.80 | 507.64 | 631.87 | 1642.90 | 687.42 |

Table 4: Fine-tuning on MultiNLI dataset. (BL denotes "BERT Large", BB denotes "'BERT base')

## 4.3 EXPERIMENT ON THE EFFICIENCY OF GPU MEMORY TRAINING

We evaluate the effectiveness of our method for mitigating bias in pre-trained models. We employ the CelebA and HateXplain datasets, with tasks as delineated in sections 4.1 and 4.2.

For the CelebA dataset, we append a fairness-aware encoder on a DINO pre-trained ViT-16 model (Caron et al., 2021). For the HateXplain dataset, we incorporate a fairness-aware encoder into pre-trained models, specifically "BERT Large" and "BERT Base". Additionally, for a fair comparison,

we append a standard ViT/ BERT encoder to the ERM model. We evaluate two approaches: 1) fine-tuning all network parameters, denoted as "-FT", and 2) training only the last encoder, denoted as "-LE". We evaluate all methods on accuracy, fairness metrics, and GPU memory consumption. The results are shown in Table 5, 6.

We observed that the ERM model maintains comparable utility even when the parameters in the backbone model are frozen. This indicates that pre-trained backbones can effectively extract meaningful features. Additionally, only training the last encoder layer demands considerably less GPU memory during training. Furthermore, training only the last encoder layer resulted in enhanced fairness outcomes. This improvement is due to the fewer parameters updated, potentially mitigating overfitting issues. Notably, Ours-LE approach achieves superior fairness results while preserving a comparable utility.

| | ERM-FT | Ours-FT | ERM-LE | Ours-LE |
|---|---|---|---|---|
| DP $\downarrow$ | $19.11 \pm 1.20$ | $18.07 \pm 0.71$ | $19.12 \pm 0.71$ | $\mathbf{17.10 \pm 0.86}$ |
| EOp $\downarrow$ | $45.18 \pm 2.77$ | $\mathbf{33.94 \pm 0.76}$ | $38.84 \pm 1.79$ | $36.33 \pm 1.96$ |
| EOd $\downarrow$ | $24.27 \pm 1.38$ | $\mathbf{18.73 \pm 0.34}$ | $20.83 \pm 1.05$ | $19.71 \pm 0.86$ |
| ACC $\uparrow$ | $\mathbf{95.59 \pm 0.09}$ | $94.05 \pm 0.22$ | $94.98 \pm 0.18$ | $94.14 \pm 0.17$ |
| GPU (MiB) $\downarrow$ | 11792 | 11926 | 2925 | 3210 |

Table 5: Comparing Fine-Tuning vs. Training the Last Encoder on the CelebA Dataset.

| | BERT Large | | | | BERT Base | | | |
|---|---|---|---|---|---|---|---|---|
| | ERM-FT | Ours-FT | ERM-LE | Ours-LE | ERM-FT | Ours-FT | ERM-LE | Ours-LE |
| DP $\downarrow$ | $12.38 \pm 2.78$ | $11.65 \pm 0.91$ | $\mathbf{10.11 \pm 2.19}$ | $12.74 \pm 1.09$ | $8.88 \pm 0.84$ | $11.30 \pm 1.97$ | $\mathbf{10.17 \pm 2.23}$ | $10.29 \pm 1.18$ |
| EOp $\downarrow$ | $5.43 \pm 3.56$ | $5.31 \pm 1.38$ | $8.27 \pm 4.68$ | $\mathbf{2.64 \pm 0.97}$ | $8.55 \pm 1.27$ | $\mathbf{4.74 \pm 3.28}$ | $6.62 \pm 1.80$ | $4.97 \pm 1.14$ |
| EOd $\downarrow$ | $5.37 \pm 1.75$ | $3.51 \pm 1.08$ | $6.67 \pm 3.35$ | $\mathbf{2.17 \pm 1.05}$ | $6.62 \pm 0.44$ | $3.40 \pm 0.97$ | $6.07 \pm 1.62$ | $\mathbf{3.24 \pm 0.78}$ |
| ACC $\uparrow$ | $\mathbf{79.69 \pm 0.15}$ | $79.61 \pm 0.32$ | $78.74 \pm 0.74$ | $76.84 \pm 0.45$ | $77.74 \pm 0.43$ | $77.89 \pm 0.78$ | $\mathbf{79.13 \pm 0.26}$ | $76.12 \pm 0.05$ |
| GPU(MiB) | 16411 | 16493 | 7022 | 7253 | 7830 | 7899 | 5303 | 5393 |

Table 6: Comparing Fine-Tuning vs. Training the Last Encoder on the HateXplain Dataset.

### 4.4 FAIRNESS-ACCURACY TRADE-OFF

We explore the effect of hyperparameters. We tune the following hyperparameters: **Ours**: We change $t$, where $t$ is the number of value vectors to perform the alignment. We choose $t \in \{2, 4, 6\}$; **KD**: We change $\lambda$ in the loss function that governs how much soft-label is being used. We choose $\lambda \in [0.3, 0.7, 1]$; **DRO**: We change $\eta$ in the loss function that controls the range of the worst-case groups. We choose $\eta \in \{0.1, 0.3, 0.5\}$; **ARL**: We change $l$ the number of encoders stacked for the adversarial network. We choose $l \in \{2, 4, 6\}$; **JTT**: We change the up-weight coefficient $\lambda_{up} \in \{20, 50, 100\}$. Inspired by (Kim et al., 2020), we depict a Pareto frontier to analyze the trade-off between fairness and accuracy. We use CelebA dataset with $a$ =male and $y$ =Blond hair. All other hyperparameters remain unchanged with Table 2. All methods share the same seed. The results can be found in Fig 2.

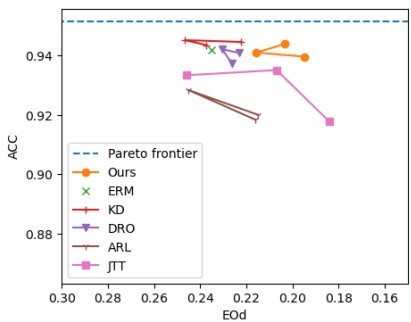

Figure 2: Effects of hyperparameters on the fairness-accuracy tradeoff.

Observations indicate that JTT outperforms in fairness performance. However, its sensitivity to hyperparameters suggests reduced robustness. Both our method and KD show comparable utility, but our approach outperforms in terms of fairness. DRO demonstrates less sensitivity to hyperparameter variations. Our method is close to the theoretical tradeoff line between fairness and accuracy, achieving an optimal balance between fairness and accuracy.

## 5 SUMMARY

In this paper, we introduce a novel method to tackle fairness issues in transformers without demographics. To mitigate bias in attention weight allocation, we normalize $\mathbf{q}$ and $\mathbf{k}$, then take their absolute values to compute the attention weight. We address bias in value vectors through local alignment. We conduct rigorous theoretical analysis to validate our method. We address fairness issues in pre-trained models and propose a method of using an additional fairness-aware encoder to mitigate these concerns. To validate our approach, we conduct experiments on various datasets in computer vision and NLP domains. Our results demonstrate the efficacy of our method in addressing fairness issues. Furthermore, we verify that our approach is useful for improving fairness in pre-trained models even with limited GPU memory resources.

ACKNOWLEDGEMENTS

This work was partially supported by the EMBRIO Institute, contract #2120200, a National Science Foundation (NSF) Biology Integration Institute, and NSF IIS #1955890, IIS #2146091.

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

## A  EMPIRICAL VALIDATION ASSUMPTION 1

We use ERM to train a ViT on the CelebA dataset. We save the model that achieves the highest validation accuracy. From the last encoder, we extract $\mathbf{q}_{\text{CLS}}$ and $\mathbf{K}_{10}$, where "CLS" refers to a special token and $\mathbf{K}_{10}$ represents the key vector of the 10-th token. We randomly select one dimension from $\mathbf{q}_{\text{CLS}}$ and $\mathbf{K}_{10}$ to generate two conditional distribution plots based on $a$.

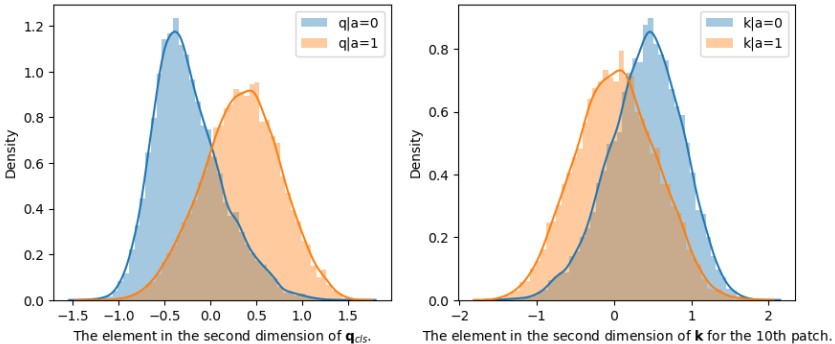

Figure 3: Histogram of a random selected dimension of $\mathbf{q}_{\text{cls}}$ and $\mathbf{k}$

## B  PROOF FOR THEROREM 1

We first consider a 1-D case and next generalize the derivation to $d_k$ dimensions.

For the 1-D derivation, now we focus only on $q$, then

$$q \sim \lambda_0 \mathcal{N}(\mu_0, \sigma_0^2) + \lambda_1 \mathcal{N}(\mu_1, \sigma_1^2) \tag{9}$$

where $\lambda_0 + \lambda_1 = 1$. Then the expectation of $q$ is

$$\mu := \mathbb{E}[q] = \lambda_0 \mu_0 + \lambda_1 \mu_1 \tag{10}$$

and the variance can be written as

$$\sigma^2 := \text{Var}[q] = \mathbb{E}[q^2] - (\mathbb{E}[q])^2, \tag{11}$$

$$= \lambda_0 \mathbb{E}_{\mathcal{N}(\mu_0, \sigma_0^2)}[q^2] + \lambda_1 \mathbb{E}_{\mathcal{N}(\mu_1, \sigma_1^2)}[q^2] - (\lambda_0 \mu_0 + \lambda_1 \mu_1)^2 \tag{12}$$

$$= \lambda_0 (\mu_0^2 + \sigma_0^2) + \lambda_1 (\mu_1^2 + \sigma_1^2) - (\lambda_0 \mu_0 + \lambda_1 \mu_1)^2 \tag{13}$$

$$= (\lambda_0 \sigma_0^2 + \lambda_1 \sigma_1^2) + \lambda_0 \lambda_1 (\mu_0 - \mu_1)^2 \tag{14}$$

$$= (\lambda_0 \sigma_0^2 + \lambda_1 \sigma_1^2) + \lambda_0 \lambda_1 \Delta^2 \quad \text{WLOG, let } \Delta = \mu_0 - \mu_1 > 0 \tag{15}$$

After normalization with $q^{norm} = \frac{q - \mathbb{E}[q]}{\sqrt{\text{Var}[q]}}$, the conditioned variable $q^{norm}|a = 0$ and $q^{norm}|a = 1$ are still Gaussian, and the conditioned expectations can be written as

$$\mu_{0,norm} := \mathbb{E}[q^{norm}|a = 0] = \frac{\mu_0 - \mu}{\sigma} = \frac{\lambda_1 \Delta}{\sqrt{(\lambda_0 \sigma_0^2 + \lambda_1 \sigma_1^2) + \lambda_0 \lambda_1 \Delta^2}} \tag{16}$$

$$\mu_{1,norm} := \mathbb{E}[q^{norm}|a = 1] = \frac{\mu_1 - \mu}{\sigma} = -\frac{\lambda_0 \Delta}{\sqrt{(\lambda_0 \sigma_0^2 + \lambda_1 \sigma_1^2) + \lambda_0 \lambda_1 \Delta^2}} \tag{17}$$

and the corresponding variances are

$$\sigma_{i,norm} = \frac{\sigma_i}{\sigma}, \quad i = 0, 1 \tag{18}$$

After taking the absolute values, that is $q^{de} = |q^{norm}|$, the conditioned expectations can be written as

$$\mathbb{E}[q^{de}|a=i] = -\int_{-\infty}^{0} tp(t)\mathrm{d}t + \int_{0}^{\infty} tp(t)\mathrm{d}t, \quad p(t) = \mathcal{N}(\mu_{i,norm}, \sigma_{i,norm}^2), i = 0,1 \quad (19)$$

$$= \sigma_{i,norm}\sqrt{\frac{2}{\pi}}\exp(-\frac{\mu_{i,norm}^2}{2\sigma_{i,norm}^2}) + \mu_{i,norm}\mathrm{erf}(\frac{\mu_{i,norm}}{\sqrt{2}\sigma_{i,norm}}) \quad (20)$$

$$(21)$$

Recall that $\mu_{i,norm} = \frac{\mu_i - \mu}{\sigma}$ and $\sigma_{i,norm} = \sigma_i/\sigma$, we have

$$\mathbb{E}[q^{de}|a=i] = \frac{\sigma_i}{\sigma}\sqrt{\frac{2}{\pi}}\exp(-\frac{(\frac{\mu_i-\mu}{\sigma})^2}{2(\frac{\sigma_i}{\sigma})^2}) + \frac{\mu_i-\mu}{\sigma}\mathrm{erf}(\frac{\frac{\mu_i-\mu}{\sigma}}{\sqrt{2}\frac{\sigma_i}{\sigma}}) \quad (22)$$

$$= \frac{1}{\sigma}\Big[\sigma_i\sqrt{\frac{2}{\pi}}\exp(-(\frac{\mu_i-\mu}{\sqrt{2}\sigma_i})^2) + (\mu_i-\mu)\mathrm{erf}(\frac{\mu_i-\mu}{\sqrt{2}\sigma_i})\Big] \quad (23)$$

$$= \frac{1}{\sigma}\Big[\sigma_i\sqrt{\frac{2}{\pi}}\exp(-(\frac{\lambda_{1-i}\Delta}{\sqrt{2}\sigma_i})^2) + (\lambda_{1-i}\Delta)\mathrm{erf}(\frac{\lambda_{1-i}\Delta}{\sqrt{2}\sigma_i})\Big] \quad (24)$$

which is because $\mu_1 - \mu = \mu_1 - \lambda_1\mu_1 - \lambda_0\mu_0 = \lambda_0(\mu_1 - \mu_0) = -\lambda_0\Delta$ and $\mu_0 - \mu = \mu_0 - \lambda_1\mu_1 - \lambda_0\mu_0 = \lambda_1(\mu_0 - \mu_1) = \lambda_1\Delta$. And $(-x)\mathrm{erf}(-x) = x\mathrm{erf}(x)$.

Note that $\frac{\lambda_{1-i}\Delta}{\sqrt{2}\sigma_i} > 0$, we have $\exp(-(\frac{\lambda_{1-i}\Delta}{\sqrt{2}\sigma_i})^2) < 1$ and $\mathrm{erf}(\frac{\lambda_{1-i}\Delta}{\sqrt{2}\sigma_i}) < 1$. As a result, the expectation can be bounded by

$$\mathbb{E}[q^{de}|a=i] \leq \frac{1}{\sigma}\Big[\sigma_i\sqrt{\frac{2}{\pi}} + \lambda_{1-i}\Delta\Big] \quad (25)$$

$$= \frac{\sigma_i\sqrt{\frac{2}{\pi}}}{\sqrt{(\lambda_0\sigma_0^2 + \lambda_1\sigma_1^2) + \lambda_0\lambda_1\Delta^2}} + \frac{\lambda_{1-i}\Delta}{\sqrt{(\lambda_0\sigma_0^2 + \lambda_1\sigma_1^2) + \lambda_0\lambda_1\Delta^2}} \quad (26)$$

$$\leq \lim_{\sigma_i\to\infty}\frac{\sigma_i\sqrt{\frac{2}{\pi}}}{\sqrt{(\lambda_0\sigma_0^2 + \lambda_1\sigma_1^2) + \lambda_0\lambda_1\Delta^2}} + \lim_{\Delta\to\infty}\frac{\lambda_{1-i}\Delta}{\sqrt{(\lambda_0\sigma_0^2 + \lambda_1\sigma_1^2) + \lambda_0\lambda_1\Delta^2}} \quad (27)$$

$$= \sqrt{\frac{2}{\pi\lambda_i}} + \sqrt{\frac{\lambda_{1-i}}{\lambda_i}} \quad (28)$$

Similar results hold for $\mathbf{k}$, too. Now the original statement can be re-written as

$$|\mu_{q_0,de}\mu_{k_0,de} - \mu_{q_1,de}\mu_{k_1,de}| \quad (29)$$

$$= \big|\mathbb{E}[q^{de}|a=0]\mathbb{E}[k^{de}|a=0] - \mathbb{E}[q^{de}|a=1]\mathbb{E}[k^{de}|a=1]\big| \quad (30)$$

$$\leq \mathbb{E}[q^{de}|a=0]\mathbb{E}[k^{de}|a=0] + \mathbb{E}[q^{de}|a=1]\mathbb{E}[k^{de}|a=1] \quad (31)$$

$$\leq \Big(\sqrt{\frac{2}{\pi\lambda_0}} + \sqrt{\frac{\lambda_1}{\lambda_0}}\Big)^2 + \Big(\sqrt{\frac{2}{\pi\lambda_1}} + \sqrt{\frac{\lambda_0}{\lambda_1}}\Big)^2 \quad (32)$$

Now consider the high-dimensional scenario, we have

$$|\delta^{de}| = \left| \mathbb{E}[\frac{\mathbf{q}^{de^\top}\mathbf{k}^{de}}{\sqrt{d_k}}|a=0] - \mathbb{E}[\frac{\mathbf{q}^{de^\top}\mathbf{k}^{de}}{\sqrt{d_k}}|a=1] \right| \tag{33}$$

$$= \frac{1}{\sqrt{d_k}} \left| \mathbb{E}[\sum_{j=1}^{d_k}(q_j^{de}k_j^{de})|a=0] - \mathbb{E}[\sum_{j=1}^{d_k}(q_j^{de}k_j^{de})|a=1] \right| \tag{34}$$

$$= \frac{1}{\sqrt{d_k}} \left| \sum_{j=1}^{d_k}\mathbb{E}[q_j^{de}k_j^{de}|a=0] - \sum_{j=1}^{d_k}\mathbb{E}[q_j^{de}k_j^{de}|a=1] \right| \tag{35}$$

$$= \frac{1}{\sqrt{d_k}} \left| \sum_{j=1}^{d_k}(\mathbb{E}[q_j^{de}k_j^{de}|a=0] - \mathbb{E}[q_j^{de}k_j^{de}|a=1]) \right| \tag{36}$$

$$\leq \frac{1}{\sqrt{d_k}} \sum_{j=1}^{d_k} \left| \mathbb{E}[q_j^{de}k_j^{de}|a=0] - \mathbb{E}[q_j^{de}k_j^{de}|a=1] \right| \tag{37}$$

$$= \frac{1}{\sqrt{d_k}} \sum_{j=1}^{d_k} \left| \mathbb{E}[q_j^{de}|a=0]\mathbb{E}[k_j^{de}|a=0] - \mathbb{E}[q_j^{de}|a=1]\mathbb{E}[k_j^{de}|a=1] \right| \tag{38}$$

$$\leq \sqrt{d_k} \left[ (\sqrt{\frac{2}{\pi\lambda_0}} + \sqrt{\frac{\lambda_1}{\lambda_0}})^2 + (\sqrt{\frac{2}{\pi\lambda_1}} + \sqrt{\frac{\lambda_0}{\lambda_1}})^2 \right]. \square \tag{39}$$

## C IMPLEMENTATION DETAILS

We summarize the implementation details as follows:

- **ERM**: We take the transformer with 8 stack encoders as the baseline model. Each encoder has 8 attention heads. We implement the ERM model using the Huggingface library without pre-trained weights. We resize the image from CelebA and UTK datasets to $64 \times 64$, and divide it into 16 patches. For CelebA and UTK, we take the AdamW as the optimizer with a learning rate of $10^{-4}$, and no scheduler is applied for the fair comparison. For NLP tasks, we take the AdamW as the optimizer with a learning rate of $10^{-5}$. We share all methods with the same configuration as the ERM model.

- Distributionally robust optimization (**DRO**): We adapt the backbone to the same as the ERM model. We tune the hyper-parameter $\eta$ at the validation set to achieve the highest accuracy. For CelebA and UTK experiments, we set $\eta = 0.15$ and $\eta = 0.10$ respectively. For HateXplain and MultiNLI, we set $\eta = 0.25$.

- Adversarially reweighted learning (**ARL**): We adapt the learner network to the same as the ERM model. For the adversarial network, we apply a 6-layer stack encoder in the transformer. This is a smaller configuration compared to the learner network, as recommended by the authors.

- Fairness without demographics through knowledge distillation (**KD**): We adapt the student model that is the same as the ERM model. For the teacher network, we follow the suggestion of the authors that use a larger network, hence we adopt the vision transformer with 12 stacked encoder layers. The student network is trained by the output of the teacher model with softmax activation.

- Just train twice (**JTT**): We adapt the backbone network is the same as the ERM model. For CelebA, we follow the suggestion in the paper choose the number of epochs of training the identification model $T = 1$. During the second training, we choose the upsampling factor $\lambda_{\text{up}} = 50$. For UTK, we set $T = 10$, $\lambda_{up} = 50$. For HateXplain, we set $T = 20$, $\lambda_{up} = 50$. For MultiNLI, we set $T = 2$, $\lambda_{up} = 20$.

- Learning from failure (**LfF**): We take the networks for a biased model and a debiased model are the same as the ERM model. We set the amplification coefficient in generalized cross entropy loss $q = 0.7$ as suggested in the original paper.

- **Ours**: For vision tasks, our method employs a backbone identical to the ERM model, which comprises an 8-stacked encoder vision transformer. In the final encoder, we normalize and apply the absolute value to $\mathbf{q}$ and $\mathbf{k}$ for each head. From this encoder layer, we select the $\mathbf{v}$ vectors associated with the two highest attention weights and execute a local alignment. For NLP tasks, we extend a pre-trained model with an additional encoder layer, applying our method specifically to this added layer. Similarly, in the NLP tasks, we choose the $\mathbf{v}$ vectors with the two highest attention weights for alignment.

## D    EVALUATION METRICS

The group fairness metrics are measurements of the performance of different sensitive groups. We focus on three specific metrics: Demographic Parity, Equal Opportunity, and Equalized Odds. Demographic Parity (DP): DP focuses on the equality of the outcomes across different demographic groups, regardless of their abilities. It ensures that each group receives positive outcomes at the same rate. Equal Opportunity (EOp): EOp ensures that samples who should receive a positive outcome have an equal chance of being correctly identified, regardless of their group. Equalized Odds (EOd): EOd requires both that samples that should receive a positive outcome have an equal chance of being correctly identified (like EOp), and also that samples that should receive a negative outcome have an equal chance of being correctly identified, across all sensitive groups. The computations for the fairness metrics are as follows:

$$
\begin{aligned}
\text{DP} = |PP_i - PP_j|, \quad & \text{EOp} = |TPR_i - TPR_j|, \\
\text{EOd} = \frac{1}{2}(|TPR_i - TPR_j| + |FPR_i - FPR_j|), \quad & i, j \in \mathcal{A}
\end{aligned}
\tag{40}
$$

where $PP, TPR$, and $FPR$ are the positive prediction rate, the true positive rate, and the false positive rate.

## E    ATTENTION WEIGHT VISUALIZATION

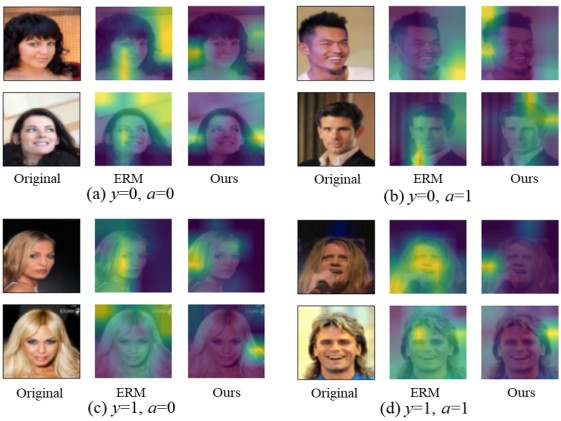

Figure 4: Visualization of attention weight. $y =$ blond hair, $a =$ male in CelebA.

We present a visualization of the average attention weight in the last encoder layer for both the ERM and our proposed models in Fig 4. We opted for models that demonstrated the highest validation accuracy. The ERM model achieved an accuracy of 94.27%, our model achieved an accuracy of 94.08%. We observe inconsistencies in attention allocation using the ERM training objective function. Despite its high accuracy, the ERM model predominantly focuses on facial features, failing to distribute attention adequately. In contrast, our model provides a more uniform attention allocation, effectively reducing the focus on facial features or other irrelevant features.

## F    ABLATION STUDY

We perform an ablation experiment from three perspectives to understand their importance. **w/o local alignment**: We train our model without incorporating the local value alignment technique. The training process is solely guided by the cross-entropy loss. **w/o debias attention**: While training our model, we exclude the normalization and exclude the absolute values in vectors $\mathbf{q}$ and $\mathbf{k}$ when calculating the attention weight. **w/o absolute value**: We train our model using normalized vectors $\mathbf{q}$ and $\mathbf{k}$ to compute the attention weight, but we exclude applying the absolute value on these vectors. We use CelebA dataset with $a =$male and $y =$Blond hair. All the methods share the same seed with the Baseline model.

| Method | DP↓ | EOp ↓ | EOd ↓ | ACC ↑ |
|---|---|---|---|---|
| Baseline | 16.99 | 43.04 | 23.05 | 94.20 |
| w/o local alignment | 18.70 | 39.59 | 21.72 | 93.94 |
| w/o debias attention | 16.59 | 43.36 | 22.92 | **94.63** |
| w/o absolute value | 18.24 | 37.44 | 20.59 | 94.39 |
| Ours | **15.81** | **36.49** | **19.47** | 93.96 |

Table 7: Ablation study on CelebA dataset.

Table 7 demonstrates the efficacy of our design modules. Notably, without the debias attention, the outcomes are close to those of the Baseline model. This highlights the significance of debias attention as an essential component for debiasing attention. Concurrently, the local value alignment technique further improves fairness, with a large impact on EOd. When the network integrates both techniques, it achieves optimal fairness with a slight drop in accuracy.

## G    ALGORITHM

We summarize our training algorithm in Algo 1.

## H    LIMITATION

In the realm of both computer vision and natural language processing, the transformer architecture stands as a pivotal framework. Our study introduces a method specifically designed for transformers, avoiding the use of sensitive information. We recognize that our approach's primary limitation lies in its application to binary-sensitive attributes, a simplification compared to the multifaceted nature of real-world scenarios. Expanding our methodology to accommodate arbitrary sensitive attributes represents a significant and promising avenue for future research.

---

**Algorithm 1** Debias Attention mechanism

---

**Input:** Input tokens $\mathbf{X} \in \mathbb{R}^{N \times d_{\text{model}}}$, label $\mathbf{y}$.
Weight matrices $\mathbf{W_Q} \in \mathbb{R}^{d_{\text{model}} \times d_k}$, $\mathbf{W_K} \in \mathbb{R}^{d_{\text{model}} \times d_k}$, $\mathbf{W_V} \in \mathbb{R}^{d_{\text{model}} \times d_v}$ for Query, Key, Value.
**Output:** The loss of local alignment on value

1: Project $\mathbf{X}$ to Query, Key, and Value matrices: $\mathbf{Q} \leftarrow \mathbf{XW_Q}$, $\mathbf{K} \leftarrow \mathbf{XW_K}$, $\mathbf{V} \leftarrow \mathbf{XW_V}$
2: For each $\mathbf{q} \in \mathbf{Q}$, $\mathbf{k} \in \mathbf{K}$:
$$\mathbf{q}^{de} = m \circ n(\mathbf{q}), \ \ \mathbf{k}^{de} = m \circ n(\mathbf{k})$$

$m$ and $n$ are defined in Theorem 1.
3: Compute attention weights $w_i$:
$$w_i = \frac{\mathbf{q}^{\text{de}}_{\text{cls}} \cdot \mathbf{k}^{\text{de}}_i}{\sqrt{d_k}}$$

4: Let $W = \{w_1, w_2, ..., w_l\}$ where $l$ is the number of tokens. Select value vectors corresponding to the top $t$ attention weights:
$$\mathbf{s}^* = \arg\max_{\mathbf{s}} \sum_{i \in \mathbf{s}} w_i \text{ s.t. } w_i \in W, |\mathbf{s}| = t$$

5: Extract and concatenate the top $t$ value vectors:
$$\mathbf{v} \leftarrow \text{concat}(V_{\mathbf{s}^*})$$

6: Mapping to the contrastive space:
$$\mathbf{z} = g(\mathbf{v})$$

7:
$$\mathcal{L}_{\text{alignment}} = \sum_i \frac{-1}{|P(i)|} \sum_{p \in P(i)} \log \frac{\exp(\mathbf{z}_i \cdot \mathbf{z}_p / \tau)}{\sum_{k \in B(i)} \exp(\mathbf{z}_i \cdot \mathbf{z}_k / \tau)},$$

where $i \in I = \{1, ..., n\}$, $B(i) = I \backslash \{i\}$, $P(i) = \{p \in B(i) : y_p = y_i\}$, $\tau = 0.07$.
 **return** $\mathcal{L}_{\text{alignment}}$

---

