# OpenReview forum: "Debiasing Attention Mechanism in Transformer without Demographics"
_ICLR.cc/2024/Conference — ICLR 2024 poster_

### Official Review · Reviewer_DJw2 · 2023-10-31

**Soundness:** 3 good
**Presentation:** 3 good
**Contribution:** 2 fair
**Rating:** 6
**Confidence:** 4

**Summary:**

The paper aims at the fairness issue when deploying models. This paper presents an approach to debiasing transformers using their inherent structure. The authors propose some methods to handle the queries, keys and values to reduce the bias. Also, the memory efficiency in the training phase is enhanced.

**Strengths:**

1. The overall writing is clear.
2. The problem of fairness issue when deploying models is important.

**Weaknesses:**

The biggest problem is the experiment settings and results.

1. In the experiment tables, which method belongs to "Fairness without Demographics", which method requires Demographics?
2. The proposed method cannot beat SOTA methods. For example, in Table 2, the proposed method is worse than LfF on EOp dataset. In Table 4, the proposed method is worse than JTT method on EOp and EOd dataset.
3. Following problem 2, the authors may claim that they have much less energy consumption. However, the comparison is not straightforward. The authors need to show one of the two results to claim this point: 1) same energy consumption and higher accuracy; 2) same accuracy and less energy consumption. If the authors can show these results compared to LfF on EOp dataset, and JTT method on EOp and EOd dataset. I will raise my score.
4. Energy consumption is not a stable indicator when comparing the models. The hardware may have big influence. I would recommend to use FLOPs to compare these methods.

**Questions:**

see weakness

---

> ### Author Response · Authors · 2023-11-20
> **To Reviewer Djw2**
>
> The authors would like to thank to the reviewer's for thorough and careful reviewing our work. We appreicate the insightful feedback and suggestions provided. To address the concerns raised, we offer the following clarifications:
>
> >**Q1**: Which method belongs to "Fairness without Demographics", which method requires Demographics?
>
> **Response:** We note that for all the methods we compared, none of those utilize demographics in training procedures. We provide detailed illustration on implementation of all methods in Appendix Section C.
>
> >**Q2**: Comprision with SOTA methods.
>
> **Response:** We recognize that our method does not outperform all other methods across every dataset. As shown in Table 2, Learning from Failure (LfF) attains better Equality of Opportunity (EOp) results, albeit at the cost of a roughly $6\%$ decrease in accuracy compared to the Empirical Risk Minimization (ERM) model. We also observe that the average accuracy of $87.97\%$ is only marginally higher than the class rate, where $p(y=0) = 84.62\%$ in the test set. However, our method achieves comparable utility to the ERM model while enhancing group fairness.
>
> In Table 4, as we discussed in Section 4.2, the significant disparity between the rates $p(a=0)=92.9\%$ and $p(a=1)=7.1\%$ leads to an increased theoretical upper bound. As specified in Theorem 1, where $λ_0=p(a=0)$ and $λ_1=p(a=1)$, this upper bound is directly proportional to $\frac{λ_0}{λ_1}$. This disparity accounting for the performance drop in this test. Although Just Train Twice (JTT) shows superior fairness outcomes, it requires considerably more computational resources compared to other baseline methods.
>
>
> >**Q3**:
> Experiments on the energy controlled setting.
>
> **Response:** In line with the reviewer's recommendation, we have conducted additional experiments with a specific focus on energy consumption. We set a predetermined energy budget and, within this constraint, trained various methods to observe their performance. The other experimental settings remain consistent with those detailed in Table 2 and Table 4 (in the paper). To ensure a fair comparison, we established the energy budget at the mean level of energy consumption observed across all methods. For the CelebA dataset, we allocate a power energy of 363.63 Wh, and for MultiNLI dataset, we allocate a power energy of 2742.78 Wh for BERT large, 867.46 Wh for BERT base. The results could be found in Table 1, 2, and 3.
>
> Table 1: Results of the MultiNLI Experiment under a Limited Energy Budget of 2742.78 Wh (Backbone: BERT Base).
>
> |  | DP ↓    | EOp ↓ | EOd ↓  | ACC ↑ | Energy   |
> |--------|--------|-------------|-------------|------|-----|
> | JTT  | 47.42 | 12.54| 11.37 | 83.37| 2313.27
> | Ours  | 41.19| 10.15| 8.12| 83.96| 1903.87
>
>
> Table 2: Results of the MultiNLI Experiment under a Limited Energy Budget of 867.46 Wh (Backbone: BERT Base).
>
> |  | DP ↓    | EOp ↓ | EOd ↓  | ACC ↑ | Energy   |
> |--------|--------|-------------|-------------|------|-----|
> | JTT  | 46.28 | 14.63 | 13.22 | 78.88 | 766.92
> | Ours  | 46.94 | 13.27 | 12.33| 81.38| 684.42
>
>
> Table 3: Results of the CelebA Experiment under a Limited Energy Budget of 363.63 Wh.
>
> |  | DP ↓    | EOp ↓ | EOd ↓  | ACC ↑ | Energy   |
> |--------|--------|-------------|-------------|------|-----|
> | LfF  | 21.07 | 45.48 |25.92 |92.67 | 258.65
> | Ours  | 14.26| 39.70 | 20.84 | 93.59 | 175.23
>
> Tables 1, 2, and 3 indicate that, under a limited energy budget, our method not only achieves improved utility but also yields enhanced fairness results simultaneously. In contrast, under the same energy constraints, the early stopping for JTT impedes its ability to achieve fair results. For LfF (Learning from Failure), the reduced number of iterations improves its utility. However, this comes at the cost of its fairness performance.
>
> >**Q4**:
> Hardware influence on Enery metrics.
>
> **Response:**
> (1) We would like to clarify that in this study, we conducted controlled experiments, ensuring that all methods were executed under identical configurations. The number of Floating Point Operations (FLOPs) is used to quantify the total count of elementary machine operations. In line with the reviewer's recommendation, we employed FLOPs as the metric to evaluate all methods. We carry out the experiment on CelebA dataset, with $y=$Blond Hair and $a=$Male. We use the *thop* library to estimate the total training FLOPs.
>
> Table 4: Comparative Analysis of FLOPs Required by Different Methods During the Training Stage
>
> |  | ERM | DRO | ARL | KD| JTT| LfF| Ours
> |-------|---------|----|-----|-------------|-------------|------------|-------------|
> | FLOPs  | $3.10×10^{13}$ | $3.10×10^{13}$ | $5.14×10^{13}$ |  $7.74×10^{13}$ | $1.34×10^{14}$| $8.26×10^{13}$|$4.13×10^{13}$|

---

> > ### Comment · Reviewer_DJw2 · 2023-11-21
> >
> > Thanks for your effort in responding.
> > I have a question about the setting in the experiment in the response. Why did the authors choose JTT for the MultiNLI Experiment and LfF for the CelebA Experiment? Does the JTT provide a better energy/accuracy trade-off on MultiNLI than other methods such as *Distributionally robust optimization (DRO) (Hashimoto et al., 2018), Adversarially reweighted learning
> > (ARL) (Lahoti et al., 2020), Fairness without demographics through knowledge distillation (KD)
> > (Chai et al., 2022), and Learning from failure (LfF)*. If not, please compare the best method that has the best energy/accuracy trade-off.

---

> > > ### Author Response · Authors · 2023-11-22
> > > **Response to reviewer DJw2**
> > >
> > > Thanks for your question. The models listed in our response were chosen because they outperform our method in terms of fairness metrics without an energy budget. We conduct comprehensive experiments within the energy budget. The energy budget is determined based on the average energy consumption observed across all implemented methods.  For the CelebA dataset, we tested the methods under a 363.63 Wh energy budget. The results are as follows:
> > >
> > > Table 1: Results of the CelebA Experiment under a Limited Energy Budget of 363.63 Wh.
> > >
> > > |  | DP ↓    | EOp ↓ | EOd ↓  | ACC ↑ |
> > > |--------|--------|-------------|-------------|------|
> > > |DRO|16.62 | 42.81| 22.88| 94.04|
> > > |ARL| 17.61 | 46.32| 24.89| 94.43|
> > > |KD| 17.10 | 42.75| 22.84| 94.24|
> > > |JTT|14.86 | 43.87| 23.33| 93.54|
> > > | LfF  | 21.07 | 45.48 |25.92 |92.67 |
> > > | Ours  | 14.26| 39.70 | 20.84 | 93.59 |
> > >
> > > For the NLP tasks, as we mentioned in the second paragraph in section 4.2, the utilization of a pre-trained model is essential. Methods that require an auxiliary network, such as ARL, KD, and LfF, are incompatible with the specifications of a pre-trained model. Hence, we exlude those methods in comparison. For NLP tasks, the results are shown in Tables 2 and 3.
> > >
> > > Table 2: Results of the MultiNLI Experiment under a Limited Energy Budget of 2742.78 Wh (Backbone: BERT Large).
> > >
> > > |  | DP ↓    | EOp ↓ | EOd ↓  | ACC ↑ |
> > > |--------|--------|-------------|-------------|------|
> > > | DRO| 47.25 | 12.11 | 10.40|83.93|
> > > JTT  | 47.42 | 12.54| 11.37 | 83.37|
> > > | Ours  | 41.19| 10.15| 8.12| 83.96|
> > >
> > >
> > > Table 3: Results of the MultiNLI Experiment under a Limited Energy Budget of 867.46 Wh (Backbone: BERT Base).
> > >
> > > |  | DP ↓    | EOp ↓ | EOd ↓  | ACC ↑ |
> > > |--------|--------|-------------|-------------|------|
> > > | DRO |47.76| 14.73| 12.74| 81.76|
> > > | JTT  | 46.28 | 14.63 | 13.22 | 78.88 |
> > > | Ours  | 46.94 | 13.27 | 12.33| 81.38|
> > >
> > > The results show that within the allocated energy budget, our method achieves optimal fairness while maintaining utility comparable to other baseline models across the three settings.

---

> > > > ### Comment · Reviewer_DJw2 · 2023-11-22
> > > >
> > > > Thanks for the results. Although the accuracy is not as good as DRO and ARL, other metrics show improvement.
> > > >
> > > > I will raise my score to 6.

---

### Official Review · Reviewer_mDe1 · 2023-11-06

**Soundness:** 3 good
**Presentation:** 3 good
**Contribution:** 3 good
**Rating:** 6
**Confidence:** 3

**Summary:**

This paper proposes a new method to address fairness issues in vision transformers and natural language processing transformers without requiring access to sensitive demographic attributes during training. The key contributions are:

- They identify two sources of bias in transformers: misallocation of attention weights and bias in value vector representations.

- To address attention weight bias, they normalize and take the absolute value of query and key vectors before computing attention. This is motivated by theoretical analysis showing it reduces disparity in attention weights between groups.

- For value vector bias, they use a supervised contrastive loss on the core value vectors to encourage consistency between groups.

- The method is evaluated on vision and NLP tasks, showing improved fairness metrics compared to prior work without demographics. It also enables efficiently debiasing pretrained models by only retraining the last encoder layer.

- Overall, the method provides a simple and effective way to improve transformer fairness without needing sensitive attributes, auxiliary networks, or restrictive constraints. The ablation studies demonstrate tradeoffs between fairness and accuracy.

**Strengths:**

- The paper tackles an important problem - improving fairness in transformers without needing sensitive attributes. This is challenging but highly relevant given privacy regulations.

- The approach of debiasing attention weights and value vectors specifically is novel. Prior work either operates on the full representations or relies on adversarial training, which can be unstable. Deconstructing the transformer in this way is creative.

- The method is simple, leveraging existing operations like normalization and contrastive loss. Avoiding complex auxiliary networks is a plus for efficiency.

- Results on vision and NLP datasets demonstrate improved fairness over prior art like DRO and knowledge distillation. The method also enables efficient debiasing of pretrained models.

- Theoretical analysis provides justification for the attention weight debiasing, and empirically shows the approach achieves near optimal fairness-accuracy tradeoffs.

**Weaknesses:**

- For NLP, the scheme of picking top value vectors may be less effective for long sequences. Dynamic selection based on attention may work better.

- The last layer retraining is convenient but provides no guarantees. Analyzing how bias propagates through the full network could further improve this.

- The contrastive loss operates locally on values. Extending the alignment more globally could potentially improve fairness further without sacrificing accuracy.

- No rigorous ablation study is provided to analyze the individual effects of the attention and value debiasing components. Their relative contributions are unclear.

**Questions:**

see weakness

---

> ### Author Response · Authors · 2023-11-20
> **To Reviewer mDe1**
>
> We are grateful for the reviewer's dedicated and thorough review of our work, as well as their constructive suggestions for its improvement. We would like to address the following concerns raised:
>
> >**Q1**: For NLP, the scheme of picking top value vectors may be less effective for long sequences. Dynamic selection based on attention may work better.
>
> **Response:** We agree with the reviewer's opinion that dynamic selection could be an effective mechanism for long sequences.  In our current method, we select the top $k$ vectors based on attention weights and concatenate these vectors. Subsequently, these concatenated vectors are transformed into the contrastive space using a nonlinear function $g(⋅)$, which is implemented via a Multilayer Perceptron (MLP). However, due to the constrain of the fixed length input of MLP models, incorporating dynamic selection (may vary the number of concatenated vectors) can be very challenging. We consider it a valuable area for future exploration.
>
> >**Q2**:
> The last layer retraining is convenient but provides no guarantees. Analyzing how bias propagates through the full network could further improve this.
>
> **Response:** For the last layer retraining we integrate our debiasing encoder with a pre-trained model. During the training phase, we freeze the pre-trained model, focusing on training the debiasing encoder and classifier for the downstream task. As the pre-trained model remains frozen during training,  biases inherent in its representations are unavoidable. Our proposed method functions as a debiasing encoder, designed to minimize sensitive information in the final representation. For details, we kindly direct the reviewer to our general response.
>
>
> >**Q3**:
> The contrastive loss operates locally on values. Extending the alignment more globally could potentially improve fairness further without sacrificing accuracy.
>
> **Response:** Extending contrastive loss to include all patches is effectively the same as applying it to the representation $\mathbf{z}$. However, it's important to note that directly aligning the representation $\mathbf{z}$ does not necessarily contribute to fairness. Park et al [1] demonstrate that with the use of Supervised Contrastive Learning (SupCon) to pre-train a ResNet model, unfairness issue still exists.  We attribute this phenomenon to the correlation between the target and sensitive labels. Since the network contrasts the entire representation, it can still leverage sensitive-related information.
>
> In contrast, our proposed method involves two steps. The first is debiasing the attention weight, a crucial step as it plays a role in selecting high attention patches. The second step involves using screened local values for contrastive learning, focusing on target-related information, thereby enhancing fairness. Our ablation study, detailed in section F of the Appendix, reveals that employing local value alignment alone has a limited impact on promoting fairness.
>
>
> >**Q4**:
> Missing ablation study
>
> **Response:** Kindly refer to our Appendix, we provide an ablation study in section F.
>
> References:
>
> - [1] Park, S., Lee, J., Lee, P., Hwang, S., Kim, D., & Byun, H. (2022). Fair contrastive learning for facial attribute classification. In Proceedings of the IEEE/CVF Conference on Computer Vision and Pattern Recognition (pp. 10389-10398).

---

### Official Review · Reviewer_sKv2 · 2023-11-09

**Soundness:** 2 fair
**Presentation:** 3 good
**Contribution:** 3 good
**Rating:** 5
**Confidence:** 4

**Summary:**

This work focuses on the attention mechanism to debias the transformers without assuming access to the sensitive attribute information. For this, two steps have been performed. First, they propose a weight relocation mechanism by normalizing (subtracting mean and dividing by standard deviation) and taking the absolute value of query ank key vectors in the attention module.
A theoretical insight is also provided to show that this bounds the discrepancy for various sensitive attributes.
Then, a nonlinear mapping is applied on tokens with higher attention values to map them to a latent representation $v \rightarrow z$. Then, supervised contrastive learning is applied to the latent representation to make sure that the embedding for the samples from the same class has core similarity.

**Strengths:**

The paper is generally written well and easy to follow, even though some parts are missing (will discuss later).

The idea of focusing on the attention module and debiasing the transformer without assuming access to the sensitive attributes is really interesting.

As this approach is not computationally intensive (based on the experimental results provided in section 4) and is not making significant changes to the architecture, it can be easily added during training to most of the transformer-based structures and improve fairness.

**Weaknesses:**

This paper has some weaknesses and I believe addressing these weaknesses can improve the quality of the paper:

1. Intuitively I understand that as the attention mechanism shows the importance of different patches during training/ inference, it can have a large impact on introducing bias. However, authors need to justify in a more systematic way, why using only the attention mechanism is powerful enough to debias a transformer structure. As an example, an analysis in a controlled setup (when we have access to sensitive attributes) can be provided to show that the attention module is the one that mostly affects the bias, or similar experiments to justify this.

2. Similarly, authors need to justify why the optimization objective in Eqn. (2) (minimizing the disparity in attention weights) can be a good approximation to the fairness metrics? I believe there should be an approximation error as instead of considering the output, we are just considering the attention weights. A detailed analysis is required to justify this alternative definition.

3. The statistics of the $q$ and $k$ are estimated during training and then used as an estimation during inference. Authors should provide more details on the accuracy of this choice, as the distribution shift during training and inference might affect both fairness and classifier performance.

4. In section 3.4., more details are required regarding debiasing the pre-trained network by inserting the encoder layer. This part was not clear to me.

5. The experimental results are not convincing enough as a very limited number of combinations for label $y$ and sensitive attributes $A$ are used.
- Authors should provide various combinations to show the generalizability of the results.
- In addition, for different datasets, different combinations are used which may give a bad impression of cherry picking.
- On average, the proposed method is not better than previous approaches and the main benefit is less compute. I wonder wether this approach can be combined with previous methods to give better performance in terms of reducing bias and preventing the degradation in the classifier accuracy?

small typo in Figure 1: dotted line $\rightarrow$ solid line

I am willing to increase my score if the authors provide proper responses.

**Questions:**

please refer to weaknesses

---

> ### Author Response · Authors · 2023-11-20
> **To Reviewer sKv2 (Part 1)**
>
> The authors would like to thank to reviewer sKv2 for your careful review and constructive suggestions. We would like to address the following questions:
>
>
> >**Q1 & Q2**: Justify debiasing only on the attention mechanism is powerful enough. Why the optimization objective in Eqn. (2) can be a good approximation to the fairness metrics?
>
>
> **Response:** Please kindly refer to our general response Q1 for details.
>
>
> >**Q3**: The difference between the training set estimated statistics and test set statistics. Also concern about distribution shift.
>
>
> **Response:** Following the reviewer's suggestion, we run the experiment on CelebA, with the task $y$=Blond hair, $a=$Male. We report $||\mathbb{E}[\mathbf{Q}]||_F$, $||\mathbb{E}[\mathbf{K}]||_F$, $||\sigma[\mathbf{Q}]||_F$ and $||\sigma[\mathbf{K}]||_F$ in the test set.  We also report running mean, and running standard deviation estimated in the training set.
>
>
>
> Table 2: Statistics in the test set and estimated statistics in the training set
>
> | $eq$ | $eq_r$ | $diff_{eq}$ ↓ | $ek$ | $ek_r$ | $diff_{ek}$↓  |
> |--------|----------------|--------------|--------------|--------------------------|-------------|
> | 143.91| 142.86 | 23.60 | 180.39 | 176.39  | 14.92  |
>
> where $eq$ = $||\mathbb{E}[\mathbf{Q}]||_F$,
>
> $eq_r= ||\mathbf{e_{\text{running mean for query}}}||_F$,
>
> $diff_{eq}$ = $||\mathbb{E}[\mathbf{Q}]-\mathbf{e}_{\text{running mean for query}}||_F$,
>
> $ek = ||\mathbb{E}[\mathbf{K}]||_F$,
>
> $ek_r$= $||\mathbf{e}_{\text{running mean for key}}||_F$,
>
> $diff_{ek}$ = $||\mathbb{E}[\mathbf{K}]-\mathbf{e}_{\text{running mean for key}}||_F$
>
> | $sq$ | $sq_r$ | $diff_{sq}$ ↓ | $sk$ | $sk_r$ | $diff_{sk}$↓  |
> |--------|----------------|--------------|--------------|--------------------------|-------------|
> | 231.22 | 233.05| 14.13 | 167.23  | 157.03  | 25.01  |
>
> $sq = ||\sigma[\mathbf{Q}]||_F$,
>
> $sq_r$= $||\mathbf{s}_{\text{running std for query}}||_F$,
>
> $diff_{sq}$= $||\sigma[\mathbf{Q}]-\mathbf{s}_{\text{running std for query}}||_F$,
>
> $sk$ = $||\sigma[\mathbf{K}]||_F$,
>
> $sk_r$= $||\mathbf{s}_{\text{running std for key}}||_F$,
>
> $diff_sk$ = $||\sigma[\mathbf{K}]-\mathbf{s}_{\text{running std for key}}||_F$.
>
>
>
> It is important to highlight that in practice, we can only estimate the statistics of the test set based on the running statistics from the training set, a common approach also utilized in batch normalization and layer normalization. Table 3 is presented as a reference to evaluate the quality of this estimation. Our observations from Table 3 indicate that the distribution shift in the CelebA dataset, for the task $y=$Blond Hair, $a=$Male, is relatively minor. We acknowledge that significant deviations between the statistics estimated by the running mean and standard deviation and those in the test set will have an impact on our method's performance, potentially leading to a decreased utility. In fact, the distribution shift is also a challenging problem in the ML community.
>
> >**Q4**: Clarification is needed on the  "last encoder training" section.
>
> **Response:** We please reviewer kindly refer to our Q2 in the general response.

---

> ### Author Response · Authors · 2023-11-20
> **To Reviewer sKv2 (Part 2)**
>
> >**Q5**:
> (1). Limited evalutaion. (2). Concern about task selection (3). Compatibility of existing work.
>
> **Response:**
> - Following the reviewer's suggestion, we have carried out additional experiments on the CelebA dataset, which is notable for its 40 attributes and widespread use in fairness research. Specifically, we focused on tasks commonly explored in fairness studies. For the CelebA dataset, the tasks are $y=$ Wavy Hair $a=$ male [1] and $y=$ Bags under eyes $a=$ Young [2]. The results can be found in Tables 3 and 4.
> - For different datasets, we do not pick a particular task. Instead, we followed the experiments outlined in [3, 4, 5], which are widely recognized in fairness research.
> - We provide an additional experiment to demonstrate the compatibility of our method. We conducted tests combining our approach with the Just Train Twice (JTT), the results are in Table 5. We thank the reviewer for the suggestion and will add full results to our final paper.
>
> Table 3: Classification results on CelebA dataset: $a$= male, $y$ = Wavy hair.
>
> |    | ERM    | DRO    | ARL   | KD    | JTT  | LfF*  | Ours           |
> |--------|----------|---------|--------|-----------|------|------|---------|
> | DP ↓  | 30.76 ± 1.51   | 34.85 ± 1.43   | 27.67 ± 1.18   | 29.91 ± 3.77   | 29.80 ± 1.45    |   -  | 28.96 ± 2.74   |
> | EOp↓         | 40.52 ± 1.80   | 44.63 ± 2.62   | 40.54 ± 1.44   | 41.26 ± 1.13   |  29.16 ± 0.44    |  -  | 34.93 ± 0.51   |
> | EOd ↓        | 28.14 ± 1.38   | 31.79 ± 1.82   | 27.26 ± 1.23   | 28.02 ± 2.07   | 24.23 ± 0.96     |  -  | 24.82 ± 1.32   |
> | ACC ↑        | 74.11 ± 0.14   | 73.97 ± 0.32   | 72.11 ± 0.04   | 73.46 ± 0.86   |  69.87 ± 0.36  |   63.60*  | 73.26 ± 0.38   |
>
> *LfF makes all predictions to the same target group. We did a grid search on the learning rate from \{$10^{-3}, 10^{-4}, 10^{-5}, 10^{-6}$\} with optimizer \{$Adam, AdamW$\}.
>
> Table 4: Classification results on CelebA dataset: $a$ = young, $y$ = Bags under eyes.
>
> |              | ERM            | DRO            | ARL            | KD             | JTT           | LfF  | Ours           |
> |--------------|----------------|----------------|----------------|----------------|---------------|------|----------------|
> | DP ↓         | 15.70 ± 1.13   | 9.13 ± 0.50    | 7.92 ± 0.85    | 10.83 ± 1.49   | 10.57 ± 0.23  |   8.95 ± 2.46  | 6.34 ± 0.97    |
> | EOp↓         | 15.03 ± 1.68   | 6.15 ± 2.35    | 6.97 ± 1.08    | 6.98 ± 1.47    | 2.27 ± 0.88   |  7.75 ± 2.81    | 4.10 ± 0.50    |
> | EOd ↓        | 11.56 ± 0.72   | 5.35 ± 1.07    | 5.65 ± 0.78    | 6.18 ± 0.92    | 4.71 ± 0.54   |  6.23 ± 2.06    | 3.70 ± 0.41    |
> | ACC ↑        | 83.15 ± 0.13   | 82.49 ± 0.38   | 81.57 ± 0.24   | 82.96 ± 0.23   | 78.78 ± 0.76  |   81.66 ± 0.47   | 81.80 ± 0.52   |
>
>
> Table 5: Compatible experiment on CelebA: $a$ = Male, $y$ = Blond Hair.
>
> |    | ERM   |   JTT  |Ours| Ours+JTT  |
> |----------|----------------|----------------|---------------|-----------|
> | DP ↓  | 16.92 ± 0.55  | 18.04 ± 0.32 | 16.34 ± 1.94 | 16.91 ± 0.45  |
> | EOp↓   | 42.82 ± 0.47  | 36.17 ± 1.08 | 38.01 ± 1.32   | 28.77  ± 2.44   |
> | EOd ↓  | 22.90 ± 0.10 | 20.26 ± 1.72  | 20.43 ± 0.68  |  16.47  ± 1.07   |
> | ACC ↑  | 94.26 ± 0.06 | 92.75 ± 0.02 | 94.06 ± 0.43  |   91.83  ± 0.91   |
>
> Analysis of Tables 3 and 4 shows that our method outperforms other baseline models in various tasks. While Just Train Twice (JTT) maintains good fairness, it experiences a slight drop in accuracy. Interestingly, as Table 5 illustrates, combining our method with JTT leads to enhanced outcomes. This improvement is attributed to JTT's ability to modify the distribution among sensitive groups, thereby balancing the data distribution across different groups and potentially reducing the unfairness upper bound as outlined in Theorem 1. Moreover, our method's use of contrastive learning techniques further enhances the utility of the JTT approach.
>
> References:
> - [1] Han, Xiaotian, et al. "FFB: A Fair Fairness Benchmark for In-Processing Group Fairness Methods." arXiv preprint arXiv:2306.09468 (2023).
> - [2] Park, Sungho, et al. "Fair contrastive learning for facial attribute classification." Proceedings of the IEEE/CVF Conference on Computer Vision and Pattern Recognition. 2022.
> - [3] Liu, Evan Z., et al. "Just train twice: Improving group robustness without training group information." International Conference on Machine Learning. PMLR, 2021.
> - [4] Hong, Youngkyu, and Eunho Yang. "Unbiased classification through bias-contrastive and bias-balanced learning." Advances in Neural Information Processing Systems 34 (2021): 26449-26461.
> - [5] Baldini, Ioana, et al. "Your fairness may vary: Pretrained language model fairness in toxic text classification." arXiv preprint arXiv:2108.01250 (2021).

---

### Official Review · Reviewer_JY1q · 2023-11-10

**Soundness:** 4 excellent
**Presentation:** 2 fair
**Contribution:** 3 good
**Rating:** 6
**Confidence:** 4

**Summary:**

**Rebuttal Update** After reading the author's rebuttal, I improved my score from a 5 to a 6. However, I would like to see the writing and clarity improve for a final paper if accepted. Specifically, the introduction and use of notation needs cleaning, as well as the motivations in the introduction. I would also like a bit more explanation of the contrastive learning method and its motivations in the context of the whole work.

In this paper, the authors propose a new method for debiasing the attention mechanism to achieve fairness without prior subgroup definitions. Their method consists of two components. First, they normalize the token embeddings in their Query and Key matrices and absolute value them to bound the attention weight difference across sensitive attributes. Second, they use a contrastive loss to encourage the embeddings of samples from the same class to be similar to each other, encouraging equal representation across sensitive attributes while maintaining performance. Then, the authors provide extensive empirical evaluation of their fairness method across two vision and two language tasks.

**Strengths:**

This paper builds on a wide literature of debiasing methods, and does a great job of synthesizing methods from fairness aware transformers with methods in fairness without demographics. Furthermore, the method is straightforward and the first part can be applied easily to any attention mechanism, whereas the second requires a small amount of fine tuning of a nonlinear head. The entire system can be applied out of the box to existing models with little or no training. Another thing I really liked about the paper is the green analysis of the power consumption of the method. I think this is a great step forward for researchers in the field.

**Weaknesses:**

One weakness of the paper is the motivation. While debiasing models / fairness is generally a strong motivation, the authors do not explain in much detail why it is important to debias attention mechanisms. Instead, most of the introduction feels like a "related works" section, where the main motivation of the paper is the failings of previous methods. I would like to see a better written introduction that explains why the failings of previous methods are bad or costly, and why a new method is needed.

Other parts of the paper are not very clear as well. What motivates the fairness optimization problem (2)? Why do we care about attention weight values when in fairness we traditionally care about outcomes? While having the same activations results in the same outputs of course, it seems to me to be extreme to limit the expressivity of the model across sensitive attributes.

Furthermore, the notation in the paper is pretty messy and unclear. In section 3.1, in the first paragraph the vectors $\textbf{q}, \textbf{k}$ are not introduced as slices of the $Q$ and $K$ matrices. Also the relationship between the dataset $\mathcal{D}$ and how it is inputted into the attention mechanism/transformer model is not mentioned at all (it just jumps straight from dataset notation to attention notation with no connection between the two). In Section 3.2 near the end, what is $q_{cls}$?

I would like to see a better figure explaining the pipeline or mathematical equation of the entire model. $g$ is only mentioned before eqn. (7), but not ever shown visually, this makes it very unclear as to how to implement the second modification and is a bit misleading as we require an additional layer to train to align the model.

Finally, I would like to see an ablation study of the two mechanisms to compare which one impacts fairness more.

**Questions:**

What is the practical/fairness motivation behind the fairness optimization problem, and the motivation behind debiasing attention as a whole?

Also, it seems as though instead of using subgroup attributes in the contrastive loss, you use classes. However, what happens if each class is dominated by a single sensitive group attribute? For example, if class 1 is all male and class 2 is all female, then requiring all class 1 (male) representations to be similar with each other and different than class 2 (female) will actually cause more disparity by pushing the representations away from each other. Don't we have to assume sensitive groups are balanced within classes?

Finally, just as a curiosity, won't normalizing reduce the expressivity of the model? I would love to see the distribution of attention weights before and after normalizing, as well as before and after absolute valuing (as well as an ablation study of the two steps).

---

> ### Author Response · Authors · 2023-11-20
> **To reviewer JY1q (Part 1)**
>
> Thank you for carefully reviewing our work, also thank you for your valuable suggestions. We want to take this opportunity to address the following concerns:
>
> >**Q1**:
> Motivation: Why is it important to debias attention mechanisms? Why do we need a new method?
>
> **Response:** The Transformer architecture is well-known for its effectiveness, but studies [1, 2] have noted that it is affected by statistical biases. For a detailed discussion on the significance of debiasing attention mechanisms, kindly please refer to the general response to Question 1.
>
>  Current methods addressing fairness issues without demographics can be broadly divided based on the use of auxiliary networks. Methods that do not require auxiliary networks include Distributionally Robust Optimization (DRO) [10] and Just Train Twice (JTT) [9], while those employing auxiliary networks including Adversarial Representation Learning (ARL) [3], Knowledge Distillation (KD) [7], and Learning from Failure (LfF) [8].
>
> Our approach is specifically designed to address fairness without demographics and without the need for an auxiliary network, thereby reducing parameters for better efficiency. Regarding existing non-auxiliary network methods, DRO, as pointed out by [3], is likely to be affected by outliers, leading to a drop in performance. JTT, on the other hand, involves a costly double-training procedure, with the second phase incorporating a larger training dataset.
> In terms of the method for an auxiliary network, Knowledge Distillation (KD) utilizes a teacher model that is often larger than the student network performing the downstream task. Learning from Failure (LfF) alternates optimization between two networks of the same size, leading to parameter inefficiency. Although the auxiliary network in Adversarially Reweighted Learning (ARL) is smaller compared to the main network, its incorporation of adversarial training methods can introduce instability issues. In contrast, our approach obviates the need for such auxiliary networks, thereby enhancing parameter efficiency. Furthermore, it eliminates the necessity for multiple training phases, thus aligning with the demands of time efficiency.
>
> >**Q2**:
> The motivation behind Eqn (2). Does imposing constraints on attention limit the expressivity of the model?
>
> **Response:** Please refer to our general response of Q1.
>
>
> >**Q3**: Unclear notation. The relationship between input and attention mechanism needs clarification. What is $\mathbf{q}_{cls}$?
>
> **Response:** We appreciate the feedback from the reviewer. In our work, we denote matrices using bold uppercase letters and vectors with bold lowercase letters. For instance, the vector $\mathbf{q}$ represents a vector sliced from the matrix $\mathbf{Q}$.
>
> We will include additional illustrations to clarify how this vector is inputted into the attention mechanism. Specifically, each input sample first passes through an Embedding layer. Then, it goes a transformation via three linear modules, which map it to $\mathbf{Q}$, $\mathbf{K}$, and $\mathbf{V}$, respectively.
>
> For $\mathbf{q}_{cls}$, it represents the first token in the sequence, which is utilized for downstream tasks.
>
>
>
> >**Q4**:
> Lack of a detailed model illustration and clearer explanation of the function $g$.
>
>
>
> **Response:** We provide an algorithm for the entire model in the revised version. The function $g(\cdot)$ in our model represents a nonlinear projection head, a component utilized in contrastive learning [4,5]. We follow this protocol and employ $g(\cdot)$ to map the top attention vectors into the contrastive space. In our implementation, $g(\cdot)$ is a two-layer MLP with ReLU activation.
>
>
> >**Q5**:
> Lack of ablation study.
>
> **Response:** We provide an ablation study in Appendix section F.

---

> ### Author Response · Authors · 2023-11-20
> **To reviewer JY1q (Part 2)**
>
> >**Q6**: Do you have to assume sensitive groups are balanced within classes?
>
>
> **Response:** We do not pre-suppose that sensitive groups are evenly distributed within classes. The reasons are as follows: (1) If we assume sensitive groups are balanced within classes, research by [6] has shown that group unfairness can be significantly mitigated through ERM training. (2) In cases where each class primarily consists of a single sensitive group, samples from minority groups tend to have higher losses. This occurs because these samples are contrasted with those from different sensitive groups. By optimizing this loss, our proposed framework effectively mitigates the disparities in representation among different sensitive groups. (3) However, in the extreme case where class distinctions rely only on sensitive information, our method may not be effective. In such instances, other comparative methods like JTT also fail and work similarly to ERM training.
>
>
> >**Q7**: I would love to see the distribution of attention weights before and after normalizing, as well as before and after absolute valuing (as well as an ablation study of the two steps).
>
> **Response:** We provide a visual attention allocation comparison of our method and ERM in Appendix Section E. We will also provide more ablation results on the distribution of attention weight of the two steps in the revised version.
>
> References:
> - [1] Sudhakar, Sruthi, et al. "Mitigating bias in visual transformers via targeted alignment." arXiv preprint arXiv:2302.04358 (2023).
> - [2] Qiang, Yao, et al. "Fairness-aware Vision Transformer via Debiased Self-Attention." arXiv preprint arXiv:2301.13803 (2023).
> - [3] Lahoti, Preethi, et al. "Fairness without demographics through adversarially reweighted learning." Advances in neural information processing systems 33 (2020): 728-740.
> - [4] Khosla, Prannay, et al. "Supervised contrastive learning." Advances in neural information processing systems 33 (2020): 18661-18673.
> - [5] Chen, Ting, et al. "A simple framework for contrastive learning of visual representations." International conference on machine learning. PMLR, 2020.
> - [6] Idrissi, B. Y., Arjovsky, M., Pezeshki, M., & Lopez-Paz, D. (2022, June). Simple data balancing achieves competitive worst-group-accuracy. In Conference on Causal Learning and Reasoning (pp. 336-351). PMLR.
> - [7] Chai, Junyi, Taeuk Jang, and Xiaoqian Wang. "Fairness without demographics through knowledge distillation." Advances in Neural Information Processing Systems 35 (2022): 19152-19164.
> - [8] Nam, Junhyun, et al. "Learning from failure: De-biasing classifier from biased classifier." Advances in Neural Information Processing Systems 33 (2020): 20673-20684.
> -[9] Liu, Evan Z., et al. "Just train twice: Improving group robustness without training group information." International Conference on Machine Learning. PMLR, 2021.
> -[10] Hashimoto, Tatsunori, et al. "Fairness without demographics in repeated loss minimization." International Conference on Machine Learning. PMLR, 2018.

---

> > ### Comment · Reviewer_JY1q · 2023-11-21
> > **Discussion Reponse**
> >
> > Thank you for answering my questions!
> >
> > I appreciate the inclusion of ablation studies as well as empirical justification of why minimizing eqn. (2) improves fairness performance. Furthermore, I appreciate the visualization of attention weights with and without your method to understand how your method impacts performance.
> >
> > It is clear to me now that your method directly addresses issues of bias, in that attention mechanisms can be sources of unfairness in decisionmaking systems. In the final paper, I would love to see a representative Figure 1 making it clear that these attention mechanisms can be biased and your method corrects for this bias, similar to the Figure 1 of the Debiased Self Attention paper you cited in your rebuttal. You included it in the Appendix, but I think this will really help the paper's message.
> >
> > I also appreciate your justification of why using class-labels for contrastive learning does not just proxy demographic attributes. I think including your justification of this in the main paper would be useful, as it would link this second method more closely to your motivation and your overall system.
> >
> > With these technical updates, I will improve my score from a 5 to a 6. However, I still think the writing in this paper needs to be cleaned up, specifically the notation (explain the lowercase bold q, k, label q_cls, and connect the setup with the dataset to the transformer architecture section). Furthermore, I would like greater clarity in how your contrastive loss component connects to the system and is motivated by your fairness goals. Finally, I think a lengthier discussion of motivations would greatly improve the paper.

---

> > > ### Author Response · Authors · 2023-11-22
> > > **Response to reviewer JY1q**
> > >
> > > Thank you for your prompt response. Following your constructive suggestions, we have revised the paper as follows:
> > >
> > > (1) A background and notation is added to Section 3 to explain how inputs are transformed within the transformer architecture and to clarify the meanings of each variable.
> > >
> > > (2) The motivation section is revised to discuss why we focus on debiasing within the attention mechanism.
> > >
> > > (3) A paragraph is added to Sec 3.2 to explain how local alignment contributes to fairness.
> > >
> > > (4) A representative figure is incorporated into Sec 3.3 to illustrate why the ERM model may exhibit unfairness and how our method adjusts the allocation of attention weights. This figure also depicts the connection of the contrastive loss component in our work.
> > >
> > > (5) An algorithm section is added to Appendix G to delineate clearly the training process of a model utilizing our method.

---

### Author Response · Authors · 2023-11-20
****General Response** (Part 1):**

The authors thank all reviewers for their dedicated review of our work and for providing valuable suggestions.

We would like to address the following common concerns raised by the reviewers:

1. Why is debiasing on the attention mechanism sufficient?
2. What is "Last encoder training (section 3.4)"? Why can we do this?

**Response to Q1:** The Transformer architecture is renowned for its effectiveness in CV and NLP domains. However, studies [1, 2] have highlighted its suffering from statistical biases. Specifically, [1] reveals disparities in average attention across different sensitive groups under the same task, underscoring the importance of the attention mechanism in mitigating bias issues. Additionally, [1,2] have shown that addressing fairness concerns within the attention mechanism is effective and yields good performance compared to other in-processing methods.

To validate the effectiveness of debiasing the attention mechanism, *Reviewer sKv2* has recommended conducting controlled experiments to demonstrate the influence of the attention module on bias. In accordance with this suggestion, we have carried out controlled experiments to substantiate the effectiveness of debiasing within the attention mechanism.

Experimental setup: We train a vanilla ViT model in CelebA as the baseline (ERM), the task is $y$="Blond Hair" and $a$=Male. We apply Eqn (2):$\min |\delta|\: s.t. \ \delta = \mathbb{E}[w|a=0]-\mathbb{E}[w|a=1]$ as our objective to debias on attention weight. We extract the attention weights from the last encoder, and split the attention weights based on the sensitive labels. We implement Eqn (2) as a fairness regularization term, we train the ViT on the objective function: $\mathcal{L_1}=\mathcal{L_\text{ce}}+\mathcal{L_{\text{MSE}}}(\mathbb{E}[\mathbf{w}|a=0],\mathbb{E}[\mathbf{w}|a=1])$. To further substantiate our approach, we investigate debiasing the Query and Key components separately. Specifically, we extract the Query and Key from the last encoder and set the following objective:
 $\mathcal{L_2}=\mathcal{L_\text{ce}}+\mathcal{L_\text{MSE}}(\mathbb{E}[\mathbf{Q}|a=0],\mathbb{E}[\mathbf{Q}|a=1])+\mathcal{L_\text{MSE}}(\mathbb{E}[\mathbf{K}|a=0],\mathbb{E}[\mathbf{K}|a=1])$.
The results are shown in Table 1.

Table 1: Fairness regularization

|   | DP ↓    | EOp ↓  | EOd ↓   | ACC ↑   |
|-------|-------|-------|----|-----|
| ERM   | 16.92 ± 0.55| 42.82 ± 0.47| 22.90 ± 0.10| 94.26 ± 0.06|
|$\mathcal{L}_1$|9.79 ± 1.14|31.56 ± 2.27| 16.41 ± 1.24| 90.08 ± 0.61
| $\mathcal{L}_2$   | 8.63 ± 0.65 | 28.76 ± 1.15| 14.91 ± 0.64| 90.32 ± 0.17|

We can observe that the regularizer enhances group fairness, validating the effectiveness of debiasing the attention weight in achieving a fairer outcome. Furthermore, the results show that debiasing each component within the attention weights significantly reduces bias, thereby underscoring the effectiveness of our proposed method.

Debiasing the attention mechanism in a transformer encoder is not only effective but also sufficient. Within one encoder layer of a transformer, there are two sub-layers: a multi-head self-attention mechanism (MHA) and a fully connected feed-forward network (FFN) [4]. Our analysis focuses on the sufficiency of debiasing within the MHA. The MHA precedes the FFN in a series configuration. The transformation of the FFN is deterministic and does not depend on the sensitive attribute $a$. Since the output of MHA, which serves as the input for the FFN, is independent of $a$, it ensures that the output of the FFN will also be independent of $a$.

We note from Table 1 that debiasing the attention mechanism slightly reduces utility. This phenomenon can be attributed to two factors. First, there is the widely accepted trade-off between fairness and accuracy. Second, as *Reviewer JY1q* mentioned, debiasing attention may constrain the model's performance. Our analysis revealed that debiasing the attention mechanism can moderately limit expressiveness, which is supported by the highest accuracy observed without debiased attention in Table 7 of the Appendix. On the other hand, the application of local alignment on value, as detailed in Section 3.2 where we employ supervised contrastive learning to contrast patches with high attention weights, appears to enhance utility. We observe that models with debiased attention but without local alignment have the lowest accuracy in the same table (Table 7 in Appendix). This suggests that debiasing attention alone might slightly restrict the model's capabilities, though the impact is minimal. However, the incorporation of supervised contrastive loss improves utility. Additionally, our findings are consistent with those in [3], which suggests that combining cross-entropy with SuperCon leads to improved utility. Furthermore, the experimental results presented in this paper, under four different datasets, empirically demonstrate that the utility of our method is preserved and comparable to ERM training.

---

### Author Response · Authors · 2023-11-20
**General Response (Part 2):**

**Response to Q2:** In section 3.4 and section 4.3, we discuss the method of the last layer retraining. This method involves appending our proposed debiasing transformer encoder to the top of a pre-trained network. During the training phase, we freeze the pre-trained network and only train the appended debiasing encoder layer and its corresponding classifier.

We would like to answer the following questions:

(1) Why is it feasible to freeze the pre-trained model while still achieving good utility?

(2) Why is training the last encoder layer sufficient to address fairness issues?


For the first question, the pre-trained transformer model has strong expressive capabilities. Studies [5, 8] demonstrate that pre-trained networks with frozen parameters achieve high utility even when utilizing a simple linear classifier. Therefore, a pre-trained model can effectively serve as a feature extractor. The features encode information related to both $y$ and $a$.

For the second question, our work is grounded in previous research work. We highlight the findings of [6], which emphasize that fine-tuning a small fraction of a neural network is adequate to achieve fairness within the network. Similarly, Kirichenko et al. [7] also suggest that debiasing a sub-network, situated closer to the model's output, can be an effective approach to addressing fairness issues. Our proposed last-layer retraining method involves adding a debiasing layer, reducing the sensitive-related information in the representation.

**Reference:**
- [1] Sudhakar, Sruthi, et al. "Mitigating bias in visual transformers via targeted alignment." arXiv preprint arXiv:2302.04358 (2023).
- [2] Qiang, Yao, et al. "Fairness-aware Vision Transformer via Debiased Self-Attention." arXiv preprint arXiv:2301.13803 (2023).
- [3] Gunel, Beliz, et al. "Supervised contrastive learning for pre-trained language model fine-tuning." ICLR 2021.
- [4] Vaswani, Ashish, et al. "Attention is all you need." Advances in neural information processing systems 30 (2017).
- [5] Caron, Mathilde, et al. "Emerging properties in self-supervised vision transformers." Proceedings of the IEEE/CVF international conference on computer vision. 2021.
- [6] Mao, Yuzhen, et al. "Last-Layer Fairness Fine-tuning is Simple and Effective for Neural Networks." arXiv preprint arXiv:2304.03935 (2023).
- [7] Kirichenko, Polina, Pavel Izmailov, and Andrew Gordon Wilson. "Last layer re-training is sufficient for robustness to spurious correlations." arXiv preprint arXiv:2204.02937 (2022).
- [8] He, Kaiming, et al. "Momentum contrast for unsupervised visual representation learning." Proceedings of the IEEE/CVF conference on computer vision and pattern recognition. 2020.

---

### Meta-Review · Area_Chair_8DAz · 2023-12-11

**Metareview:**

The manuscript introduces an approach to "debias" the attention mechanism in transformers, aimed at achieving fairness without using a "sensitive attribute" during deployment. The proposed method first employs a weight relocation mechanism by normalizing and taking the absolute values of the query and key vectors in the attention module (including a normalization step). This step has some theoretical grounding (Theorem 1). The approach then applied supervised contrastive learning to a latent representation derived from tokens with higher attention values. This ensures that embeddings of samples from the same class are closely aligned, promoting similar representations across sensitive attributes while preserving overall performance. The proposed method is tested in vision and language classification tasks.

Overall, the paper has some interesting aspects. Given the excitement around transformer architectures, the proposed approach can be used in emerging vision and text classification tasks, and does move the needle in terms of numerical benchmarks.

However, the paper also has **several** limitations, most of which were noted by the reviewers. The method is limited to binary classification tasks and, though not explicitly mentioned in the text, binary sensitive attributes --- at least all experiments and Theorem 1 are tailored to binary $a$. Though the debiasing method is also tailored to transformers, it is benchmarked on binary classification tasks for which there are **many** alternative group-fairness interventions. These alternative methods could be evaluated by the authors and are completely missed, including post-processing techniques that can be applied to any classifier and methods that aim to learn fair representations. Examples include:

* Zemel, Rich, Yu Wu, Kevin Swersky, Toni Pitassi, and Cynthia Dwork. "Learning fair representations." In International conference on machine learning, pp. 325-333. PMLR, 2013.
* Lowy, A., Pavan, R., Baharlouei, S., Razaviyayn, M., & Beirami, A. (2021). Fermi: Fair empirical risk minimization via exponential Rényi mutual information.
* Wei, D., Ramamurthy, K. N., & Calmon, F. P. (2021). Optimized score transformation for consistent fair classification. The Journal of Machine Learning Research, 22(1), 11692-11769.
* Agarwal, Alekh, Alina Beygelzimer, Miroslav Dudík, John Langford, and Hanna Wallach. "A reductions approach to fair classification." In International conference on machine learning, pp. 60-69. PMLR, 2018.
* Alghamdi, W., Hsu, H., Jeong, H., Wang, H., Michalak, P., Asoodeh, S., & Calmon, F. (2022). Beyond Adult and COMPAS: Fair multi-class prediction via information projection. Advances in Neural Information Processing Systems, 35, 38747-38760.

I strongly suggest the authors incorporate the references above (and related literature) in their paper, and articulate **why a fairness intervention tailored to transformer architectures is necessary**, particularly given that benchmarks are given in binary classification tasks -- for which all interventions above apply.

Having noted these limitations, and given the comments by the reviewers, my overall opinion of the manuscript is still somewhat favorable. Despite its limitations, the submission still moves the needle in terms of benchmarks and designing group fairness interventions tailored to transformers. Nevertheless, I strongly encourage the authors to revise their manuscript if accepted, and see the above changes as mandatory.

**Justification For Why Not Higher Score:**

The paper has several limitations, particularly in terms of writing clarity and related literature, that hold it back from a higher score.

**Justification For Why Not Lower Score:**

The paper does provide interesting benchmarks and designs a fairness intervention targeted to transformers.

---

### Decision · Program_Chairs · 2024-01-16

Accept (poster)